# Linear Transformer Topological Masking with Graph Random Features

**Isaac Reid**[*12]**, Avinava Dubey**[2]**, Deepali Jain**[3]**, Will Whitney**[3]**, Amr Ahmed**[2]**,
**Joshua Ainslie**[3]**, Alex Bewley**[3]**, Mithun George Jacob**[3]**, Aranyak Mehta**[2]**,
**David Rendleman**[3]**, Connor Schenck**[3]**, Richard E. Turner**[1]**, René Wagner**[3]**,
**Adrian Weller**[14]**, Krzysztof Choromanski**[†35]
[1]University of Cambridge, [2]Google Research, [3]Google DeepMind, [4]Alan Turing Institute,
[5]Columbia University

## Abstract

When training transformers on graph-structured data, incorporating information about the underlying topology is crucial for good performance. *Topological masking*, a type of relative position encoding, achieves this by upweighting or downweighting attention depending on the relationship between the query and keys in a graph. In this paper, we propose to parameterise topological masks as a *learnable function of a weighted adjacency matrix* – a novel, flexible approach which incorporates a strong structural inductive bias. By approximating this mask with *graph random features* (for which we prove the first known concentration bounds), we show how this can be made fully compatible with linear attention, preserving $\mathcal{O}(N)$ time and space complexity with respect to the number of input tokens. The fastest previous alternative was $\mathcal{O}(N \log N)$ and only suitable for specific graphs. Our efficient masking algorithms provide strong performance gains for tasks on image and point cloud data, including with $> 30k$ nodes.

## 1 Introduction

Across data modalities, transformers have emerged as a leading machine learning architecture (Vaswani, 2017; Dosovitskiy et al., 2020; Arnab et al., 2021). They derive their success from modelling complex dependencies between the tokens using an *attention mechanism*. However, in its vanilla instantiation the transformer is a set function, meaning it is invariant to permutation of its inputs. This is often undesireable; e.g. words tend to be arranged in a particular sequence, and graph nodes are connected to a particular set of neighbours. Therefore, it has become standard to modify the attention matrix depending on any underlying structure, building in inductive bias. For instance, *causal attention* applies a triangular binary mask to ensure that tokens only attend to preceding elements of the sequence (Yang et al., 2019). Meanwhile, *relative position encoding* (RPE) captures the spatial relationship between the tokens (Shaw et al., 2018). In some cases, the queries and keys can be arranged in a graph $\mathcal{G}$ and a function $\mathbf{M}(\mathcal{G})$ can be used to modulate the transformer attention. This is referred to as *topological masking* (Choromanski et al., 2022). It provides a powerful avenue for incorporating structural inductive bias from $\mathcal{G}$ into transformers, in some cases closing the performance gap with the best-customised graph neural networks (Ying et al., 2021).

**Topological masking of low-rank attention.** Topological masking is simple when the attention is full-rank. In this case, the entire attention matrix $\mathbf{A} \in \mathbb{R}^{N \times N}$ (with $N$ the number of input tokens) is materialised in memory, so its individual entries $\mathbf{A}_{ij}$ can be pointwise multiplied by a (fixed or learnable) mask $\mathbf{M}(\mathcal{G})_{ij}$. But full-rank attention famously incurs a quadratic $\mathcal{O}(N^2)$ space and time complexity cost, which may become prohibitive on large inputs. This has motivated a number of *low-rank decompositions* of $\mathbf{A}$. These improve scalability by avoiding computing $\mathbf{A}$ explicitly, instead doing so implicitly in feature space. The question of how to perform topological masking in the low-rank setting is challenging and not fully solved. Previously proposed algorithms are limited to inexpressive functions $\mathbf{M}$ and restricted classes of graphs $\mathcal{G}$ (e.g. trees and grids), and are often $\mathcal{O}(N \log N)$ time complexity rather than $\mathcal{O}(N)$ (App. B; Choromanski et al., 2022; Liutkus et al.,

---

[*]Work completed as a student researcher [†]Senior lead.

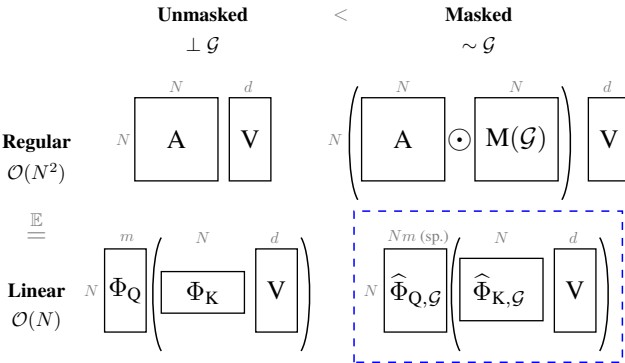

Figure 1: Schematic overview. Regular attention is $\mathcal{O}(N^2)$, with $N$ the number of input tokens. Topological masking modulates $\mathbf{A}$ by a graph function $\mathbf{M}(\mathcal{G})$, improving predictive performance. Linear attention reduces the time complexity to $\mathcal{O}(N)$ by leveraging a low-rank decomposition. Our contribution (blue) is *the first algorithm to achieve both* – $\mathcal{O}(N)$ topological masking of low-rank attention – by approximating $\mathbf{M}(\mathcal{G})$ with graph random features (GRFs). GRFs are sparse vectors (denoted 'sp.') computed by sampling random walks, constructed so that $\mathbb{E}(\widehat{\mathbf{\Phi}}_{\mathbf{Q},\mathcal{G}} \widehat{\mathbf{\Phi}}_{\mathbf{K},\mathcal{G}}^\top) = \mathbf{A} \odot \mathbf{M}(\mathcal{G})$ with strong concentration properties (Thm. 3.2).

2021). Addressing these shortcomings is the chief focus of this manuscript. Our central question is: *how can we implement efficient yet flexible topological (graph-based) masking of transformer attention, in the low-rank setting where $\mathbf{A}$ is never explicitly instantiated?*

**Graph random features.** To answer the question posed above, we propose to use *graph random features* (GRFs) (Choromanski, 2023; Reid et al., 2024b). GRFs are sparse, randomised vectors whose dot product gives an unbiased estimate of arbitrary functions of a weighted adjacency matrix, a class that captures the structure of $\mathcal{G}$ and is popular in classical graph-based machine learning (Smola and Kondor, 2003; Kondor and Lafferty, 2002). GRFs have been used for approximate kernelised node clustering and in scalable graph-based Gaussian processes (Reid et al., 2023a; 2024c), but never before in transformers. Advancing understanding of the convergence properties of GRFs is another key goal of this work.

**Our contributions.** We present the first known $\mathcal{O}(N)$ algorithm for topological masking of linear attention transformers with general $\mathcal{G}$, using graph random features (Fig. 1).

1. We propose to modulate the transformer attention mechanism with a *learnable function of a weighted adjacency matrix* – a simple, general topological masking trick which brings a strong inductive bias and substantial improvements to predictive performance.

2. To scale, we replace the exact, deterministic mask with a sparse, stochastic estimate constructed using *graph random features* (GRFs). We use a novel exponential concentration bound, requiring only a mild technical assumption on $\mathcal{G}$ as $N$ grows, to prove that our method is linear time and space complexity with respect to input size: the first known algorithm with this property.

3. We demonstrate strong accuracy gains on image data, as well as for modelling the dynamics of massive point clouds ($> 30$k particles) in robotics applications where efficiency is essential.

## 2 PRELIMINARIES

**Masked attention.** Denote by $N$ the number of input tokens and $d$ the dimensionality of their latent representations. Consider matrices $\mathbf{Q}, \mathbf{K}, \mathbf{V} \in \mathbb{R}^{N \times d}$ whose rows are given by the query ($\{\boldsymbol{q}_i\}_{i=1}^N$), key ($\{\boldsymbol{k}_i\}_{i=1}^N$) and value vectors ($\{\boldsymbol{v}_i\}_{i=1}^N$) respectively. The *attention mechanism*, a basic processing unit of the transformer, can be written:

$$\text{Att}(\mathbf{Q}, \mathbf{K}, \mathbf{V}) \coloneqq \mathbf{D}^{-1} \mathbf{A} \mathbf{V}, \quad \mathbf{A} \coloneqq \mathcal{K}(\mathbf{Q}, \mathbf{K}), \quad \mathbf{D} \coloneqq \mathbf{A} \mathbf{1}_N. \quad (1)$$

Here, the function $\mathcal{K}$ assigns a similarity score to each query-key pair and $\mathbf{1}_N \coloneqq [1]_{i=1}^N \in \mathbb{R}^N$. For regular softmax attention, one simply takes $\mathcal{K}(\mathbf{Q}, \mathbf{K})_{ij} = \exp(\boldsymbol{q}_i^\top \boldsymbol{k}_j)$, optionally also normalising by $d$. To incorporate *masking*, we make the modification

$$\mathbf{A_M} \coloneqq \mathbf{M} \odot \mathcal{K}(\mathbf{Q}, \mathbf{K}), \quad (2)$$

where $\mathbf{M} \in \mathbb{R}^{N \times N}$ is a (hard or soft) *masking matrix* and $\odot$ denotes the Hadamard (element-wise) product. For softmax, this is equivalent to adding a bias term $b_{ij} := \log \mathbf{M}_{ij}$ to $\boldsymbol{q}_i^\top \boldsymbol{k}_j$ before exponentiating. As remarked in Sec. 1, this can be used to enforce causality or as a relative position encoding (RPE) (Shaw et al., 2018). Taking $\mathbf{M}(\mathcal{G})$ so that attention is modulated by a function of the underlying graph structure is called *topological masking*. Topological masking is *not* just a mechanism to sparsify attention for efficiency gains; its goal is to incorporate topological signal from the underlying graph, improving predictive performance via a useful inductive bias.

**Improving efficiency with low-rank attention.** The time-complexity of Eq. 1 is $\mathcal{O}(N^2)$, which famously limits the ability of regular transformers to scale to long inputs. A leading approach to mitigate this and recover $\mathcal{O}(N)$ complexity is *linear* or *low-rank attention* (Katharopoulos et al., 2020; Choromanski et al., 2020). This takes a feature mapping $\phi : \mathbb{R}^d \to \mathbb{R}^m$ with $m \ll N$ and constructs the matrices $\boldsymbol{\Phi}_\mathbf{Q} := [\phi(\mathbf{q}_i)]_{i=1}^N$ and $\boldsymbol{\Phi}_\mathbf{K} := [\phi(\mathbf{k}_i)]_{i=1}^N$, where $\boldsymbol{\Phi}_{\mathbf{Q},\mathbf{K}} \in \mathbb{R}^{N \times m}$. One then defines $\mathcal{K}_{\text{LR}}(\mathbf{Q}, \mathbf{K}) := \boldsymbol{\Phi}_\mathbf{Q} \boldsymbol{\Phi}_\mathbf{K}^\top$ and computes

$$\text{Att}_{\text{LR}}(\mathbf{Q}, \mathbf{K}, \mathbf{V}) := \mathbf{D}^{-1} \left( \boldsymbol{\Phi}_\mathbf{Q} \left( \boldsymbol{\Phi}_\mathbf{K}^\top \mathbf{V} \right) \right), \quad \mathbf{D} := \boldsymbol{\Phi}_\mathbf{Q} \left( \boldsymbol{\Phi}_\mathbf{K}^\top \mathbf{1}_N \right), \tag{3}$$

where the parentheses show the order of computation. Exploiting associativity, the time complexity is reduced to $\mathcal{O}(Nmd)$, albeit usually with some sacrifice in performance compared to full-rank softmax attention. Since we care about scaling with graph size, we will henceforth write this as $\mathcal{O}(N)$. Different choices of feature maps $\phi$ exist; for example, one can take a *random feature map* that provides a Monte Carlo estimate of the softmax kernel (Performer; Choromanski et al., 2020), or a deterministic nonlinearity like $\text{elu}(x) + x$ (linear transformer; Katharopoulos et al., 2020).

**Efficient masking of low-rank attention.** As Eq. 3 suggests, low-rank attention derives its speed from avoiding computing $\mathbf{A} = \mathcal{K}_{\text{LR}}(\mathbf{Q}, \mathbf{K})$ explicitly, preferring to evaluate faster products like $\boldsymbol{\Phi}_\mathbf{K}^\top \mathbf{V}$ and $\boldsymbol{\Phi}_\mathbf{K}^\top \mathbf{1}_N$ first and use them to weight the query features $\boldsymbol{\Phi}_\mathbf{Q}$. This makes the direct Hadamard product implementation of masking in Eq. 2 impossible: if the matrix $\mathbf{A}$ is never explicitly materialised in memory, one cannot pointwise multiply each of its elements. Algorithms have been proposed to mask *implicitly* (without instantiating $\mathbf{A}$ or $\mathbf{M}$) in very simple cases, e.g. causal masking (Choromanski et al., 2020) or one-dimensional RPE (Liutkus et al., 2021; Luo et al., 2021). Subquadratic implicit masking is also possible when $\mathbf{M}$ is very structured: e.g. functions of shortest path distance on grid graphs and trees (Choromanski et al., 2022), or of the degrees of the pair of input nodes (Chen et al., 2024). See App. B for a detailed review. However, an $\mathcal{O}(N)$ algorithm capable of masking with flexible functions on general graphs has, until now, remained out of reach.

**Remainder of manuscript.** In **Sec. 3.1** we parameterise topological masks $\mathbf{M}(\mathcal{G})$ as learnable functions of weighted adjacency matrices, which incorporates strong structural inductive bias into attention and boosts performance. **Sec. 3.2** shows how this can be implemented *implicitly*, without instantiating $\mathbf{A}$ or $\mathbf{M}$ in memory. **Sec. 3.3** offers a faster stochastic alternative using GRFs, which we prove is $\mathcal{O}(N)$ time and space complexity. **Sec. 4** gives experimental evaluations across data modalities, including images and point clouds. The **Appendices** present proofs and extra details too long for the main text, and extended discussion about related work.

## 3 Towards linear topological masking

This section details our main results. Consider an undirected graph $\mathcal{G}(\mathcal{N}, \mathcal{E})$ where $\mathcal{N} := \{v_1, ..., v_N\}$ is the set of nodes and $\mathcal{E}$ is the set of edges, with $(v_i, v_j) \in \mathcal{E}$ if and only if there exists an edge between $v_i$ and $v_j$ in $\mathcal{G}$. The size of the graph $N$ corresponds to the number of tokens. Denote by $\mathbf{M}(\mathcal{G}) \in \mathbb{R}^{N \times N}$ the topological masking matrix.

### 3.1 Graph node kernels as topological masks

As remarked above, a common choice for $\mathbf{M}(\mathcal{G})_{ij}$ is a function of the shortest path distance between nodes $v_i$ and $v_j$ (Ying et al., 2021). Though effective at improving performance, this requires computation for every pair of nodes so it cannot straightforwardly be applied to low-rank attention in a scalable way – except for special structured $\mathcal{G}$ where the mask becomes simple (Choromanski et al., 2022). Shortest path distances are also very sensitive to changes like the addition or removal of edges, and need access to the whole graph upfront.

**A new topological mask: graph node kernels.** Given that we intend $\mathbf{M}(\mathcal{G})_{ij}$ to quantify some sense of topological similarity between $v_i$ and $v_j$, a more principled choice may be *graph node*

*kernels*: positive definite, symmetric functions $k : \mathcal{N} \times \mathcal{N} \to \mathbb{R}$ mapping from pairs of graph nodes to real numbers. These are widely used in bioinformatics (Borgwardt et al., 2005), community detection (Kloster and Gleich, 2014) and recommender systems (Yajima, 2006), with more recent applications in manifold learning for deep generative modelling (Zhou et al., 2020) and solving single- and multiple-source shortest path problems (Crane et al., 2017). They are also used to model the covariance function in geometric Gaussian processes (Zhi et al., 2023; Borovitskiy et al., 2021).

Graph node kernels are often parameterised as follows. Let $\mathbf{W} \in \mathbb{R}^{N \times N}$ be a *weighted adjacency matrix*, so that $\mathbf{W} \coloneqq [w_{ij}]_{i,j \in \mathcal{N}}$ with $w_{ij}$ nonzero if and only if $(v_i, v_j) \in \mathcal{E}$. A common choice is $w_{ij} = 1/\sqrt{d_i d_j}$, with $d_i$ the degree of $v_i$. Then consider the power series

$$\mathbf{M}_\alpha(\mathcal{G}) \coloneqq \sum_{k=0}^{\infty} \alpha_k \mathbf{W}^k \tag{4}$$

where $(\alpha_k)_{k=0}^{\infty} \subset \mathbb{R}$ is a set of real-valued Taylor coefficients. This is positive definite if and only if $\sum_{k=0}^{\infty} \alpha_k \lambda_i^k > 0 \, \forall \, \lambda_i \in \Lambda(\mathbf{W})$, where $\Lambda(\mathbf{W})$ denotes the set of eigenvalues of $\mathbf{W}$. With different choices for $(\alpha_k)_{k=0}^{\infty}$, Eq. 4 includes many of the most popular graph node kernels in the literature, e.g. heat, diffusion, cosine and $p$-step random walk. Kernel learning can be implemented by optimising $\alpha_k$ (Reid et al., 2024b; Zhi et al., 2023).

Eq. 4 provides a very effective parameterisation for topological attention masks, striking the right balance between flexibility and hard-coded structural inductive bias. In Sec. 4 we will show that it boosts performance compared to unmasked attention. However, explicitly computing $\mathbf{M}_\alpha(\mathcal{G}) \in \mathbb{R}^{N \times N}$ and storing it in memory is incompatible with $\mathcal{O}(N)$ low-rank attention. The next section will show how this can be avoided by masking *implicitly*.

### 3.2 IMPLICIT MASKING WITH GRAPH FEATURES

Linear attention is fast because it replaces softmax attention $\exp(\mathbf{q}_i^\top \mathbf{k}_j)$ with a 'featurised' alternative $\phi(\mathbf{q}_i)^\top \phi(\mathbf{k}_j)$. Can we do the same for $\mathbf{M}_\alpha(\mathcal{G})$? The following is true.

**Lemma 3.1** (Explicit $N$-dimensional features for graph node kernels (Reid et al., 2024b)). *Let* $(f_k)_{k=0}^{\infty} \subset \mathbb{R}$ *denote the deconvolution of* $(\alpha_k)_{k=0}^{\infty}$, *i.e.* $\sum_{p=0}^{k} f_p f_{k-p} = \alpha_k \, \forall \, k$. *There exists a feature matrix* $\mathbf{\Phi}_\mathcal{G} \in \mathbb{R}^{N \times N}$ *satisfying* $\mathbf{M}_\alpha(\mathcal{G}) = \mathbf{\Phi}_\mathcal{G} \mathbf{\Phi}_\mathcal{G}^\top$, *given by* $\mathbf{\Phi}_\mathcal{G} \coloneqq \sum_{k=0}^{\infty} f_k \mathbf{W}^k$.

This is can be shown in a few lines of algebra; see App. A.1. In general, one requires some extra technical assumptions to ensure that $\mathbf{M}_\alpha(\mathcal{G})$ and $\mathbf{\Phi}_\mathcal{G}$ converge, which we also describe in App. A.1. We refer to the rows of the feature matrix $\mathbf{\Phi}_\mathcal{G}$ as *graph features*, letting $\mathbf{\Phi}_\mathcal{G} \eqqcolon [\phi_\mathcal{G}(v_i)]_{i=1}^{N}$ with $\phi_\mathcal{G}(v_i) \in \mathbb{R}^N$ the graph feature of node $v_i$. For every kernel there trivially exists a feature map (Gretton, 2013), but unusually in this case it is finite-dimensional and admits a simple closed form.

Let $\otimes : \mathbb{R}^m \times \mathbb{R}^N \to \mathbb{R}^{m \times N}$ denote the *outer product* which maps a pair of vectors to a matrix, and $\text{vec} : \mathbb{R}^{m \times N} \to \mathbb{R}^{mN}$ denote the *vectorising* operation that flattens a matrix into a vector. Then $\phi(\mathbf{q}_i) \otimes \phi_\mathcal{G}(v_i)$ is an $m \times N$ matrix whose $jk$th element is $\phi(\mathbf{q}_i)_j \phi_\mathcal{G}(v_i)_k$, and $\text{vec}(\phi(\mathbf{q}_i) \otimes \phi_\mathcal{G}(v_i))$ is an $mN$-dimensional vector whose $(N(j-1)+k)$th entry is $\phi(\mathbf{q}_i)_j \phi_\mathcal{G}(v_i)_k$. Observe the following:[1]

$$\text{vec}(\phi(\mathbf{q}_i) \otimes \phi_\mathcal{G}(v_i))^\top \text{vec}(\phi(\mathbf{k}_j) \otimes \phi_\mathcal{G}(v_j)) = \sum_{s=1}^{mN} \text{vec}(\phi(\mathbf{q}_i) \otimes \phi_\mathcal{G}(v_i))_s \text{vec}(\phi(\mathbf{k}_j) \otimes \phi_\mathcal{G}(v_j))_s$$

$$= \sum_{k=1}^{m} \sum_{l=1}^{N} \phi(\mathbf{q}_i)_k \phi_\mathcal{G}(v_i)_l \phi(\mathbf{k}_j)_k \phi_\mathcal{G}(v_j)_l = \left(\phi(\mathbf{q}_i)^\top \phi(\mathbf{k}_j)\right) \left(\phi_\mathcal{G}(v_i)^\top \phi_\mathcal{G}(v_j)\right) = \mathcal{K}_{\text{LR}\,ij} \mathbf{M}_{\alpha\,ij}. \tag{5}$$

Hence, taking $\{\text{vec}(\phi(\{\mathbf{q}_i, \mathbf{k}_i\}) \otimes \phi_\mathcal{G}(v_i))\}_{v_i \in \mathcal{N}} \subset \mathbb{R}^{mN}$ as features, we can compute masked attention implicitly, without materialising $\mathbf{A}$ and $\mathbf{M}_\alpha$. Concretely, we have that:

$$\mathbf{\Phi}_{\mathbf{Q},\mathcal{G}} \mathbf{\Phi}_{\mathbf{K},\mathcal{G}}^\top = \mathcal{K}_{\text{LR}} \odot \mathbf{M}_\alpha \quad \text{if} \quad \mathbf{\Phi}_{\{\mathbf{Q},\mathbf{K}\},\mathcal{G}} \coloneqq [\text{vec}(\phi(\{\mathbf{q}_i, \mathbf{k}_i\}) \otimes \phi_\mathcal{G}(v_i))]_{i=1}^{N} \in \mathbb{R}^{N \times Nm}. \tag{6}$$

Masked low-rank attention can then be computed using $\mathbf{\Phi}_{\{\mathbf{Q},\mathbf{K}\},\mathcal{G}}$ as the feature matrices:

$$\text{Att}_{\text{LR},\mathbf{M}}(\mathbf{Q}, \mathbf{K}, \mathbf{V}, \mathcal{G}) \coloneqq \mathbf{D}^{-1} \left(\mathbf{\Phi}_{\mathbf{Q},\mathcal{G}} \left(\mathbf{\Phi}_{\mathbf{K},\mathcal{G}}^\top \mathbf{V}\right)\right), \quad \mathbf{D} \coloneqq \mathbf{\Phi}_{\mathbf{Q},\mathcal{G}} \left(\mathbf{\Phi}_{\mathbf{K},\mathcal{G}}^\top \mathbf{1}_N\right). \tag{7}$$

---

[1] Recall the old adage: the dot product of outer products is the product of dot products.

**Time complexity.** Eqs 6 and 7 give a recipe for computing masked attention implicitly in feature space, without needing to materialise $\mathbf{A}$ or $\mathbf{M}$ in memory. However, this approach relies on computing the feature matrix $\mathbf{\Phi}_{\{\mathbf{Q},\mathbf{K}\},\mathcal{G}} \in \mathbb{R}^{N \times Nm}$ and evaluating products like $\mathbf{\Phi}_{\mathbf{K},\mathcal{G}}^\top \mathbf{1}_N$, which still incurs $\mathcal{O}(N^2)$ space and time complexity. For a linear masking algorithm, we must reduce this to $\mathcal{O}(N)$. This can be achieved by approximating the features.

## 3.3 FASTER IMPLICIT MASKING WITH GRAPH RANDOM FEATURES

Our goal is now to replace $\mathbf{\Phi}_{\{\mathbf{Q},\mathbf{K}\},\mathcal{G}}$ by Monte Carlo estimates $\widehat{\mathbf{\Phi}}_{\{\mathbf{Q},\mathbf{K}\},\mathcal{G}}$ that satisfy

$$\mathbb{E}\left(\widehat{\mathbf{\Phi}}_{\mathbf{Q},\mathcal{G}} \widehat{\mathbf{\Phi}}_{\mathbf{K},\mathcal{G}}^\top\right) = \mathbf{\Phi}_{\mathbf{Q},\mathcal{G}} \mathbf{\Phi}_{\mathbf{K},\mathcal{G}}^\top = \mathcal{K}_{\mathrm{LR}} \odot \mathbf{M}_\alpha, \quad \widehat{\mathbf{\Phi}}_{\mathbf{K},\mathcal{G}}^\top \boldsymbol{v} \sim \mathcal{O}(N) \ \forall \ \boldsymbol{v} \in \mathbb{R}^N, \tag{8}$$

meaning we apply the desired topological mask in expectation but in linear time. This can be achieved using *graph random features* (GRFs) (Choromanski, 2023; Reid et al., 2024b).

At a high-level, GRFs can be understood as follows. From Lemma 3.1, the graph feature for node $v_i$ is given by $\phi_{\mathcal{G}}(v_i) := [\sum_{k=0}^\infty f_k \mathbf{W}_{ij}^k]_{j=1}^N$. Since $\mathbf{W}$ is an adjacency matrix, $\mathbf{W}_{ij}^k$ counts the number of graph walks of length $k$ between nodes $i$ and $j$, weighted by the product of traversed edge weights. So $\phi_{\mathcal{G}}(v_i)_j$ can be interpreted as a weighted sum over *all* walks of any length between nodes $v_i$ and $v_j$, with longer contributions suppressed since edge weights are at most 1 (by normalisation of $\mathbf{W}$). GRFs apply importance sampling, approximating this infinite sum by a Monte Carlo estimate $\widehat{\phi}_{\mathcal{G}}(v_i)$ using random walks.

**Using GRFs.** Concretely, to build the GRF $\widehat{\phi}_{\mathcal{G}}(v_i)$ one samples $n$ random walks $\{\omega_k^{(i)}\}_{k=1}^n \subset \Omega$ out of node $v_i$, where $\Omega := \left\{(v_i)_{i=1}^l \mid v_i \in \mathcal{N}, (v_i, v_{i+1}) \in \mathcal{E}, l \in \mathbb{N}\right\}$ is the set of all walks. Denote by $p(\omega)$ the probability of walk $\omega$ which is known from the sampling mechanism. $p(\omega)$ is typically chosen so that walks randomly halt, prioritising sampling lower powers of $\mathbf{W}$ that contribute more to the kernel. Let $\widetilde{\omega} : \Omega \to \mathbb{R}$ be a function computing the product of traversed edge weights,

$$\widetilde{\omega}(\omega) := \left\{\prod_{i=1}^{\mathrm{len}(\omega)} \mathbf{W}_{\omega[i]\omega[i+1]} \text{ if } \mathrm{len}(\omega) > 1, \quad 1 \text{ otherwise}\right\}, \tag{9}$$

where $\mathrm{len}(\omega)$ is the number of hops and $\omega[i]$ is the node visited at the $i$th timestep. Then:

$$\widehat{\phi}_{\mathcal{G}}(v_i)_q := \frac{1}{n}\sum_{k=1}^n \sum_{\omega_{iq} \in \Omega_{iq}} \frac{\widetilde{\omega}(\omega_{iq}) f_{\mathrm{len}(\omega_{iq})}}{p(\omega_{iq})} \mathbb{I}(\omega_{iq} \text{ prefix subwalk of } \omega_k^{(i)}), \quad q = 1,...,N, \tag{10}$$

where $\Omega_{iq}$ denotes the set of all walks between nodes $v_i$ and $v_q$, of which $\omega_{iq}$ is a member. 'Prefix subwalk' means that the walk $\omega_k^{(i)}$ beginning at node $v_i$ initially follows the sequence of nodes $\omega_{iq}$, then optionally continues. This complicated notation belies a very simple algorithm: one simulates $n$ random walks out of node $v_i$. At every visited node, one adds a contribution to the corresponding GRF coordinate that depends on 1) the product of traversed edge weights, 2) the probability of the subwalk and 3) the function $f$. The result is normalised by dividing by the number of walks $n$.

GRFs give unbiased approximation of graph node kernels: $\mathbb{E}[\widehat{\phi}_{\mathcal{G}}(v_i)^\top \widehat{\phi}_{\mathcal{G}}(v_j)] = \mathbf{M}_{\alpha ij}$. This follows because $\mathbb{E}[\mathbb{I}(\omega_{iq} \text{ prefix subwalk of } \omega_k^{(i)})] = p(\omega_{iq})$. Unbiasedness is one of our required properties; we also need $\widehat{\mathbf{\Phi}}_{\{\mathbf{Q},\mathbf{K}\},\mathcal{G}}$ to support $\mathcal{O}(N)$ matrix-vector multiplication. We will show this by providing the first known proof that GRFs are *sparse*: $\widehat{\phi}(v_i)$ has $\mathcal{O}(1)$ nonzero entries $\forall \ v_i \in \mathcal{N}$.

## 3.4 NOVEL THEORETICAL RESULTS FOR GRFs: SHARP ESTIMATORS WITH SPARSE FEATURES

It is intuitive that GRFs are sparse. The walk sampler $p(\omega)$ is chosen so that walks are short, e.g. taking simple random walks with geometrically distributed lengths. From Eq. 10, this means that $\widehat{\phi}_{\mathcal{G}}(v_i)$ is only nonzero at nodes 'close' to $v_i$. Most of its entries are 0. However, to prove that $\widehat{\mathbf{\Phi}}_{\{\mathbf{Q},\mathbf{K}\}\mathcal{G}}$ supports $\mathcal{O}(N)$ matrix-vector multiplication we need to rigorously relate this to the quality of approximation of $\mathbf{M}_\alpha(\mathcal{G})$. This requires concentration inequalities for GRFs, but no such bounds were previously known. The following result is novel.

**Theorem 3.2** (GRF exponential concentration bounds). *Consider a graph $\mathcal{G}$ with adjacency matrix $\mathbf{W}$ and node degrees $\{d_i\}_{v_i \in \mathcal{N}}$. Suppose we construct GRFs $\{\widehat{\phi}_{\mathcal{G}}(v_i)\}_{v_i \in \mathcal{G}}$ by sampling $n$ random*

walks $\{\omega_k^{(i)}\}_{v_i \in \mathcal{N}, \, k \in [\![1,n]\!]}$ with geometrically distributed lengths, terminating with probability $p_{halt}$ at each timestep. Let $c := \sum_{k=0}^{\infty} |f_k| (\max_{v_i, v_j \in \mathcal{N}} \frac{|\mathbf{W}_{ij}| d_i}{1 - p_{halt}})^k$. Suppose $c$ is finite, which is guaranteed e.g. for bounded $f$ and suitably normalised $\mathbf{W}$, or if there exists some $i_{max} \in \mathbb{N}$ such that $f_i$ vanishes for all $i > i_{max}$ (see App. A.2). Then:

$$\Pr\left(|\widehat{\phi}_{\mathcal{G}}(v_i)^\top \widehat{\phi}_{\mathcal{G}}(v_j) - \mathbf{M}_{\alpha\,ij}| > t\right) \leq 2\exp\left(-\frac{t^2 n^3}{2(2n-1)^2 c^4}\right). \tag{11}$$

*Proof sketch.* Full details are in App. A.2; here, we provide an overview. Treat each pair of random walks $X_k := (\omega_k^{(i)}, \omega_k^{(j)}) \in \Omega \times \Omega$, simulated out of nodes $v_i$ and $v_j$ respectively, as a random variable. Let $k \in [\![1,n]\!]$ so we sample $n$ such pairs of walks in total. Consider modifying one of the pairs of walks. If $c$ is finite, the maximum load it is capable of depositing is finite so we can bound the $L_1$-norm of the contribution made to $\widehat{\phi}_{\mathcal{G}}(v_i)$ and $\widehat{\phi}_{\mathcal{G}}(v_j)$. This allows us to bound the change in the estimator $\widehat{\phi}_{\mathcal{G}}(v_i)^\top \widehat{\phi}_{\mathcal{G}}(v_j)$ if $X_k$ is modified. Applying McDiarmid's inequality, the result follows. □

Thm 3.2 provides an exponential bound on the probability of a topological mask estimate deviating from its true value. Moreover, assuming that $c$ remains constant (i.e. we fix a maximum edge weight and node degree as the graph grows), this probability is *independent of the number of nodes $N$*. Therefore, if we fix $t$ and the probability $\Pr$, we can solve for the minimum number of walkers $n$ to guarantee a good mask estimate with high probability, independent of graph size. Now observe the following.

**Lemma 3.3** (GRF sparsity). *Consider a graph random feature $\widehat{\phi}_{\mathcal{G}}(v_i)$ constructed using $m$ random walkers each terminating with probability $p_{halt}$ at every timestep. With probability at least $1 - \delta$, $\widehat{\phi}_{\mathcal{G}}(v_i)$ will have $n \log(1 - (1-\delta)^{1/n}) \log(1 - p_{halt})^{-1}$ or fewer nonzero entries.*

*Proof.* Since the walk lengths are geometrically distributed, the probability that a single walk is of length $b$ or shorter is $1 - (1 - p_{halt})^b$. $n$ independent walkers are all $b$ or shorter with probability $\left(1 - (1 - p_{halt})^b\right)^n$. Let this be equal to $1 - \delta$ and solve for $b$. At most $bn$ entries of $\widehat{\phi}_{\mathcal{G}}(v_i)$ can then be nonzero, so the sparsity guarantee follows. □

Lemma 3.3 means that, if we fix the number of walkers $n$ (e.g. using the bound in Eq. 11), we can also bound GRF sparsity with high probability. Crucially: the number of nonzero GRF entries is independent of the graph size $N$. We conclude the following.

**Corollary 3.4** (GRFs implement $\mathcal{O}(N)$ topological masking). *Suppose we construct a set of graph random features $\{\widehat{\phi}_{\mathcal{G}}(v_i)\}_{v_i \in \mathcal{N}}$, choosing the number of walkers $n$ as described above with $c$, $p_{halt}$, $\delta$, $t$ and the deviation probability $\Pr\left(|\phi_{\mathrm{GRF}}(v_i)^\top \phi_{\mathrm{GRF}}(v_j) - \mathbf{M}_{\alpha\,ij}| > t\right)$ fixed as hyperparameters. Then the number of nonzero entries in $\widehat{\mathbf{\Phi}}_{\{\mathbf{Q},\mathbf{K}\},\mathcal{G}} \in \mathbb{R}^{N \times Nm}$ is linear in $N$, which means that **topological masking is implemented with $\mathcal{O}(N)$ time complexity**.*

*Proof.* Lemma 3.3 shows that the number of nonzero entries in $\widehat{\phi}_{\mathcal{G}}(v_i)$ is independent of the number of nodes $N$. This must also be true of $\mathrm{vec}(\phi(\mathbf{k}_i) \otimes \widehat{\phi}_{\mathcal{G}}(v_i))$, so the number of nonzero entries in $\widehat{\mathbf{\Phi}}_{\mathbf{K},\mathcal{G}} := [\mathrm{vec}(\phi(\mathbf{k}_i) \otimes \widehat{\phi}_{\mathcal{G}}(v_i))]_{i=1}^N$ is proportional to $N$. The same holds for the equivalent matrix for the queries, $\widehat{\mathbf{\Phi}}_{\mathbf{Q},\mathcal{G}}$. The time complexity of the sparse matrix-matrix multiplication $\widehat{\mathbf{\Phi}}_{\mathbf{K},\mathcal{G}}^\top \mathbf{V}$ is $\mathcal{O}(Nmd)$ and $\widehat{\mathbf{\Phi}}_{\mathbf{K},\mathcal{G}}^\top \mathbf{1}_N$ is $\mathcal{O}(Nm)$. Both these time complexities are linear in $N$. This is also the case for the subsequent computations involving $\widehat{\mathbf{\Phi}}_{\mathbf{Q},\mathcal{G}}$. □

**Further comments on theoretical results.** We have provided the first concentration bounds for GRFs and quantification of the tradeoff between sparsity and kernel approximation quality. These results may be of interest independent from topological masking, e.g. for applications in scalable geometric Gaussian processes (Reid et al., 2024c; Borovitskiy et al., 2021). Thm. 3.2 is **remarkably general**, requiring only the modest assumption about $\mathcal{G}$ and $f$ that $c$ is finite. To guarantee $\mathcal{O}(N)$ scaling we assume that $c$ is fixed as $\mathcal{G}$ changes, which is guaranteed by e.g. fixed maximum node degree and edge weight.

### 3.5 ALGORITHM AND INTUITIVE PICTURE

Alg. 1 presents our method. The results of Secs 3.1-3.4 can be intuitively understood as follows (see Fig. 2 for an overview). Graph node kernels, which compute an infinite weighted sum of walks

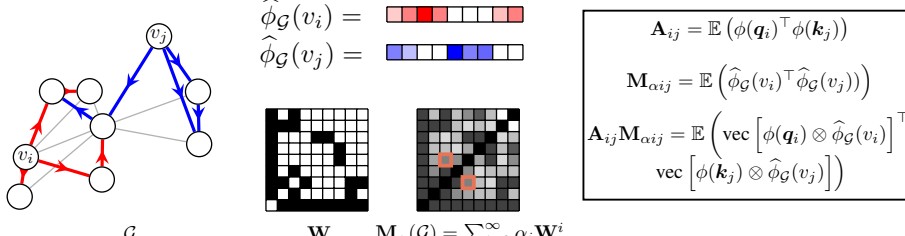

Figure 2: Visual overview. A graph $\mathcal{G}$ (left) has a weighted adjacency matrix $\mathbf{W}$ (centre left). A learnable power series $\mathbf{M}_\alpha(\mathcal{G}) := \sum_{i=0}^\infty \alpha_i \mathbf{W}^i$ is an effective topological mask or graph RPE (centre right). $\mathbf{M}_\alpha(\mathcal{G})$ can be efficiently approximated using graph random features (centre top), which perform importance sampling of halting random walks. The feature $\widehat{\phi}_\mathcal{G}(v_i)$ is only nonzero at entries visited by the ensemble of walks beginning at $v_i$. In Thm. 3.2, we prove that the number of such entries is $\mathcal{O}(1)$ whilst still accurately estimating $\mathbf{M}_\alpha(\mathbf{W})_{ij}$ with high probability, so GRFs are sparse. This unlocks $\mathcal{O}(N)$ topological masking. Note that $\widehat{\phi}_\mathcal{G}(v_i)^\top \widehat{\phi}_\mathcal{G}(v_j)$ is only nonzero if the features are nonzero at some of the same coordinates, which happens if their respective ensembles of walks 'hit'. This incorporates a strong structural inductive bias. The equations on the right formalise our method mathematically.

between pairs of input nodes, provide an effective RPE mechanism for transformers. They capture the topological relationship between queries and keys. Using graph features, we can implement this implicitly, without needing to actually materialise the mask matrix. Replacing these dense features by graph *random* features, a sparse approximation, we can reduce the cost of implicit masking to $\mathcal{O}(N)$ with low performance loss. Eq. 11 gives a remarkably tight bound on the quality of mask estimation. Physically, masking with GRFs means that nodes $v_i$ and $v_j$ only attend to one another if members of their respective ensembles of random walks *hit*, such that they visit some of the same nodes. This incorporates a strong structural inductive bias from $\mathcal{G}$. It upweights crucial interactions between nearby nodes, whilst also sampling longer-range attention with lower probability.

---

**Algorithm 1** $\mathcal{O}(N)$ topologically-masked attention for general graphs

---

**Input:** query matrix $\mathbf{Q} \in \mathbb{R}^{N\times d}$, key matrix $\mathbf{K} \in \mathbb{R}^{N\times d}$, value matrix $\mathbf{V} \in \mathbb{R}^{N\times d}$, graph $\mathcal{G}$ with weighted adjacency matrix $\mathbf{W} \in \mathbb{R}^{N\times N}$, learnable mask parameters $(f_i)_{i=0}^{i_{\max}}$, number of random walks to sample $n \in \mathbb{N}$, query/key feature map $\phi(\cdot) : \mathbb{R}^d \to \mathbb{R}^m$.
**Output:** masked attention $\mathrm{Att}_{\mathrm{LR},\widehat{\mathbf{M}}}(\mathbf{Q}, \mathbf{K}, \mathbf{V}, \mathcal{G})$ in $\mathcal{O}(N)$ time.

1: *Simulate* $n$ terminating random walks $\{\omega_k^{(i)}\}_{k=1}^n$ out of every node $v_i \in \mathcal{N}$
2: *Compute* sparse graph random features $\{\widehat{\phi}_\mathcal{G}(v_i)\}_{v_i \in \mathcal{N}} \subset \mathbb{R}^N$ using the walks $\{\omega_k^{(i)}\}_{k=1}^n$ and learnable mask parameters $(f_i)_{i=0}^{i_{\max}}$
3: *Compute* the query and key features $\{\phi(\mathbf{q}_i)\}_{i=1}^N$, $\{\phi(\mathbf{k}_i)\}_{i=1}^N$, with $\phi$ the chosen linear-attention nonlinearity (e.g. $\mathrm{elu}(x) + x$ (Katharopoulos et al., 2020) or a Monte Carlo estimate of softmax (Choromanski et al., 2020))
4: *Combine* the query/key features and graph features into a sparse topology-enhanced feature matrix, $\widehat{\mathbf{\Phi}}_{\{\mathbf{Q},\mathbf{K}\},\mathcal{G}} := [\mathrm{vec}(\phi(\{\mathbf{q}_i, \mathbf{k}_i\}) \otimes \widehat{\phi}_\mathcal{G}(v_i))]_{i=1}^N \in \mathbb{R}^{N\times Nm}$
5: *Compute* low-rank attention with topological masking in $\mathcal{O}(N)$ time by

$$\mathrm{Att}_{\mathrm{LR},\widehat{\mathbf{M}}}(\mathbf{Q}, \mathbf{K}, \mathbf{V}, \mathcal{G}) := \mathbf{D}^{-1}\left(\widehat{\mathbf{\Phi}}_{\mathbf{Q},\mathcal{G}}\left(\widehat{\mathbf{\Phi}}_{\mathbf{K},\mathcal{G}}^\top \mathbf{V}\right)\right), \quad \mathbf{D} := \widehat{\mathbf{\Phi}}_{\mathbf{Q},\mathcal{G}}\left(\widehat{\mathbf{\Phi}}_{\mathbf{K},\mathcal{G}}^\top \mathbf{1}_N\right). \quad (12)$$

---

**Distributed computation, expressivity and GATs.** Alg. 1 is simple to distribute for massive graphs that cannot be stored on a single machine; to simulate random walks one only needs a node's immediate neighbours rather than all of $\mathcal{G}$ upfront. This desirable property is not shared by masking algorithms that rely on the fast Fourier transform (Luo et al., 2021; Choromanski et al., 2022). Moreover, our method can distinguish graphs identical under the 1-dimensional Weisfeiler Lehman graph isomorphism heuristic (with colours replaced by node features) because their adjacency matrices and hence masks differ. In this sense, GRF transformers are *more expressive than standard graph*

*neural networks and unmasked transformers*, which notoriously fail this test (Morris et al., 2019; Xu et al., 2019). Finally, we remark that our algorithm can be interpreted as a stochastic extension of graph attention networks (GATs; Veličković et al., 2018), with a greater focus on linear attention and scalability. Tokens still attend to a subset of other tokens informed by the topology of $\mathcal{G}$, but importance sampling random walks permits interactions beyond nearest-neighbour. Incorporating learnable parameters $(f_i)_{i=0}^{i_{max}}$ lets us estimate a principled function of $\mathbf{W}$.

## 4 EXPERIMENTS

In this section, we test our algorithms for topological masking with GRFs. We consider data modalities with different graph topologies: images and point clouds.

### 4.1 TIME COMPLEXITY

For a hardware-agnostic comparison, we first compute the total number of FLOPs for evaluating (i) unmasked softmax, (ii) unmasked linear and (iii) GRF-masked linear attention for graphs of different sizes $N$. Fig. 3 shows the results. For concreteness we use 1-dimensional grid graphs, though the theoretical results in Sec. 3.4 are independent of $\mathcal{G}$ (provided $c$ is upper bounded). We take $n = 4$ random walkers per node with termination probability $p_{halt} = 0.5$. The latent dimension is $d = 8$ and the feature size is $m = 8$. In the masked case we show the *mean* number of FLOPs over 10 seeds since GRF sparsity is itself a random variable. As anticipated, the time complexity of our mechanism is $\mathcal{O}(N)$; there is a constant multiplicative cost for topological masking. We clearly improve upon the time complexity of $\mathcal{O}(N^2)$ softmax attention.

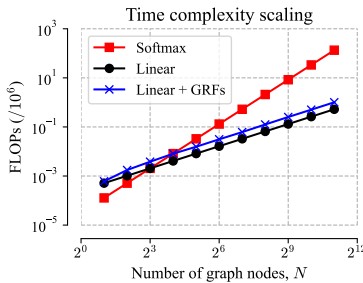

Figure 3: Number of FLOPs vs. number of graph nodes for softmax attention, linear attention, and linear attention with GRF topological masking (ours).

### 4.2 VITS: GRID GRAPHS AND IMAGE DATA

Next, we use our topological masking algorithm to improve the vision transformer (ViT; Dosovitskiy et al., 2020). The underlying graph $\mathcal{G}$ is a square grid of size `num_x_patches` $\times$ `num_y_patches`, with nodes connected by an edge if they are neighbours in $x$ or $y$. Since the graph is fixed, random walks can be pre-computed and frozen for each attention head and layer before training and inference. Full architectural and training details are in App. C. All our ViT models also endow tokens with learned additive absolute position embeddings.

Table 1 shows the final test accuracies for ImageNet (Deng et al., 2009), iNaturalist2021 (Horn et al., 2018) and Places365 (Zhou et al., 2018). For the *slow* (superlinear) variants, we include (i) unmasked softmax and (ii) Toeplitz-masked linear (Choromanski et al., 2022) attention. Toeplitz can only be used for this specific, structured graph topology. We also include (iii) $\mathbf{M}_\alpha(\mathcal{G})$-masked linear, which takes an explicit power series in $\mathbf{W}$ (Eq. 4) to directly modulate the $N \times N$ attention matrix. This is equal to our Monte Carlo GRFs algorithm in the $n \to \infty$ limit, so it indicates the asymptotic performance of GRFs with very many random walkers. For the *fast* (linear) variants,

| | Variant | Time comp. | ImageNet 1M | iNaturalist2021 2.7M | Places365 (Small) 1.8M |
|---|---|---|---|---|---|
| SLOW | Unmasked softmax | $\mathcal{O}(N^2)$ | 0.741 | 0.699 | 0.567 |
| | Toeplitz-masked linear | $\mathcal{O}(N \log N)$ | 0.733 | 0.674 | 0.550 |
| | $\mathbf{M}_\alpha(\mathcal{G})$**-masked linear** | $\mathcal{O}(N^2)$ | 0.741 | 0.692 | 0.551 |
| FAST | Unmasked linear | $\mathcal{O}(N)$ | 0.693 | 0.667 | 0.543 |
| | **GRF-masked linear** | $\mathcal{O}(N)$ | **0.730** | **0.689** | **0.548** |

Table 1: Final transformer test accuracies across attention masking variants and datasets. Our algorithms are shown in **bold**. Topological masking with GRFs is the **best** $\mathcal{O}(N)$ **mechanism**, matching or beating more expensive alternatives.

we include (iv) unmasked linear and (v) GRF-masked linear attention. Our algorithm gives strong improvements above the unmasked baseline, by $+\mathbf{3.7}\%$, $+\mathbf{2.2}\%$ and $+\mathbf{0.5}\%$ respectively. Remarkably, it is competitive with much more expensive variants: GRFs often match or beat Toeplitz. For ImageNet and iNaturalist2021, it even substantially narrows the gap with $\mathcal{O}(N^2)$ softmax.

### 4.3 ABLATION STUDIES

In App. C.2, we perform ablation studies for the ViT experiments. We find that predictive performance improves with the number of walkers $n$ because the accuracy of the Monte Carlo estimate $\widehat{\mathbf{M}}_\alpha(\mathcal{G})$ increases. Moreover, importance sampling walks is crucial; if we fail to upweight improbable walks, the gains vanish. Lastly, the inductive bias from parameterising $\mathbf{M}_\alpha(\mathcal{G})$ as a function of $\mathbf{W}$ is key. Fully learnable graph features of the same dimensionality $N$, an impractical and expensive alternative to our method, give much smaller gains.

### 4.4 PCTS: PREDICTING PARTICLE DYNAMICS FOR ROBOTICS

As a final task, we use our algorithm to model *high-density visual particle dynamics* (HD-VPD; Whitney et al., 2023; 2024), predicting the physical evolution of scenes involving a bi-manual robot using a point cloud transformer (PCT). Images from a pair of cameras are unprojected to a set of 3D particles. Their interactions, conditioned on specific robot actions, are modelled by a dynamics network. We predict updates to the latent point cloud, render the result to an image with a conditional NeRF (Mildenhall et al., 2021), and train end-to-end with video prediction loss. Such learned dynamics models are attracting growing interest in robotics as a more flexible alternative to physical simulators, but to scale to the massive point clouds needed for detailed predictions efficiency and effective structural inductive biases are key.

For the dynamics network, we use an *Interlacer*: a PCT which alternates unmasked linear attention and message passing-style updates between neighbouring particles (Whitney et al., 2024). This architecture is twice as fast as GNNs with the same prediction quality, and can handle $4\times$ bigger point clouds. To test the ability of GRFs to efficiently capture local topological structure, we replace message passing by GRF-masked linear attention, taking query/key features $\phi(\cdot) = \mathrm{ReLU}(\cdot)$. Implementation details for HD-VPD, including the robotic arm and camera configurations, are the same as reported by Whitney et al. (2024); see App. C.3 for discussion. To disambiguate, we will refer to our model as the *GRF Interlacer* and Whitney et al.'s as the *message passing (MP) Interlacer*.

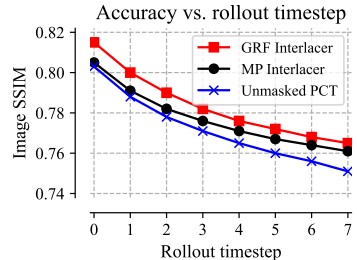

Figure 4: Accuracy comparison. Structural similarity index measure (SSIM) between ground truth camera frames and predictive NeRF renderings after 100k training steps, plotted against rollout timestep. Higher is better. GRFs improve the accuracy of dynamics predictions.

In contrast to our previous experiments, for HD-VPD the underlying graph $\mathcal{G}$ is no longer fixed so walks cannot be pre-computed. Instead, we construct them on the fly, using an approximate $k$-nearest neighbours solver to get adjacency lists of length $k = 3$ for each node. We use repelling random walks for variance reduction (Reid et al., 2024a), sampling 3 hops. For easy integration with existing Interlacer code, we use *asymmetric* GRFs (App. D; Reid et al., 2024b). We consider point clouds with $N = 32768$ particles, which is too many to explicitly instantiate $\mathbf{A}$ or $\mathbf{M}$ in memory.

Fig. 5 shows example NeRF renderings of test set predictive dynamics for four rollouts. Since the model is deterministic, predictions blur with long time horizons. Incorporating a diffusion head (Song et al., 2020) is left to future work. Fig. 4 shows predicted video frame accuracy (image SSIM) vs. rollout timestep for each model. GRFs outperform the vanilla transformer and MP Interlacer baselines, modelling the particle dynamics more faithfully. We include examples of predictive videos in the supplementary material.

In addition to modelling cloud dynamics more accurately, our method is faster. While the MP Interlacer trains at 0.925 steps/second, the GRFs Interlacer trains at 0.982 steps/second. This makes it faster by 6% – a gain our method enjoys because it avoids expensive GNN-style message passing. The unmasked baseline, which struggles to capture local structure so gives poor accuracy, trains at 1.49 steps/second.

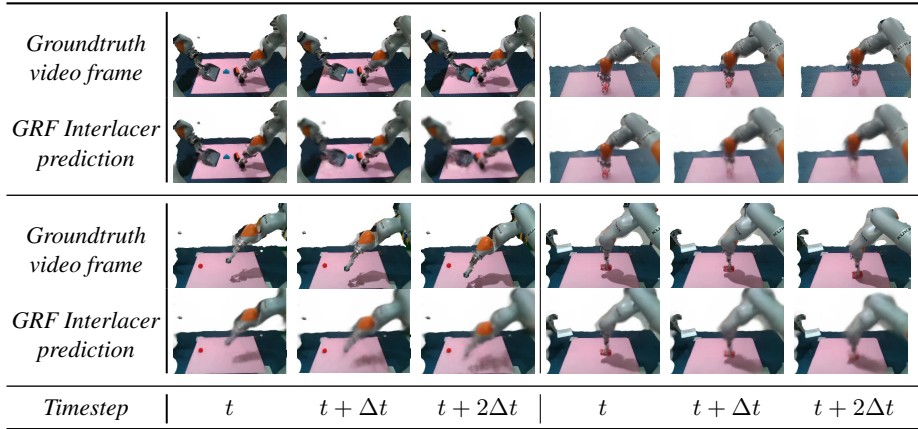

| | | $t$ | $t + \Delta t$ | $t + 2\Delta t$ | $t$ | $t + \Delta t$ | $t + 2\Delta t$ |

Figure 5: Rendered rollouts. NeRF renderings of the predictive dynamics of a bimanual Kuka robot, conditioned on the initial scene and sequence of robot actions. We show four tasks: using a dustpan and brush, lifting a can, moving a green block, and dropping a can. GRF Interlacers model point cloud dynamics more accurately, so the predicted frame rendings are closer to the ground truth.

## 5 CONCLUSION

We have presented the first linear algorithm for transformer topological masking on general graphs, based on sampling random walks. It uses graph random features, for which we have proved the first known concentration inequalities and sparsity guarantees. Our algorithm brings strong performance gains across data modalities, including on massive point clouds with $> 30$k particles for which computational efficiency is essential. Future work might include improving hardware and libraries for high-dimensional sparse linear algebra to fully benefit from the guaranteed complexity improvements, and investigating whether dimensionality reduction techniques like the Johnson-Lindenstrauss transform (Dasgupta et al., 2010) can be applied to GRFs for further speedups.

## 6 ETHICS AND REPRODUCIBILITY

**Ethics statement**: Our work is foundational with no immediate ethical concerns apparent to us. However, increases in scalability provided by improvements to efficient transformers could exacerbate existing and incipient risks of machine learning, from bad actors or as unintended consequences.

**Reproducibility statement**: We have made every effort to ensure the work's reproducibility. The core algorithm is presented clearly in Alg. 1. Theoretical results are proved with accompanying assumptions in Sec. 3.4 and App. A.2. Proof sketches are also included for clarity. Apart from the robotics dataset, all datasets are standard and freely available online. Exhaustive experimental details about the training and architectures are reported in App. C; see especially Table 2.

## 7 CONTRIBUTIONS AND ACKNOWLEDGEMENTS

**Relative contributions.** IR and KC conceptualised the project and co-designed the algorithms. IR proved the theoretical results, implemented the GRFs code, ran the Interlacer experiments and wrote the text. KAD was crucially involved throughout, running the ViT and ViViT experiments and guiding the project's direction. DJ and WW helped set up the Interlacer experiments. AA, JA, AB, MJ, AM, DR, CS, RT, RW and AW provided thoughtful feedback on the project and manuscript, helped set up experimental infrastructure, and made Cambridge and Google great places to work.

**Acknowledgements and funding.** IR acknowledges support from a Trinity College External Studentship and a Google PhD Fellowship. This work was completed whilst employed as a student researcher at Google. IR sincerely thanks Silvio Lattanzi for making this possible. RET is supported by the EPSRC Probabilistic AI Hub (EP/Y028783/1). AW acknowledges support from a Turing AI fellowship under grant EP/V025279/1 and the Leverhulme Trust via CFI.

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

# A PROOFS

## A.1 PROOF OF LEMMA 3.1

This result was presented by Reid et al. (2024b); a short demonstration is included below for the reader's convenience. Observe that

$$
\mathbf{M}_\alpha(\mathcal{G})_{ij} := \sum_{k=0}^\infty \alpha_k \mathbf{W}_{ij}^k \overset{(1)}{=} \sum_{v=1}^N \sum_{k=0}^\infty \alpha_k \mathbf{W}_{iv}^{k-p} \mathbf{W}_{vj}^p \overset{(2)}{=} \sum_{v=1}^N \sum_{k=0}^\infty \sum_{p=0}^k f_{k-p} f_p \mathbf{W}_{iv}^{k-p} \mathbf{W}_{vj}^p
$$
$$
\overset{(3)}{=} \sum_{v=1}^N \left( \sum_{k_1=0}^\infty f_{k_1} \mathbf{W}_{iv}^{k_1} \right) \left( \sum_{k_2=0}^\infty f_{k_2} \mathbf{W}_{jv}^{k_2} \right)^\top \overset{(4)}{=} \mathbf{\Phi}_\mathcal{G} \mathbf{\Phi}_\mathcal{G}^\top
\tag{13}
$$

where $\mathbf{\Phi}_\mathcal{G} := \sum_{k=0}^\infty f_k \mathbf{W}^k$. Here, step (1) splits the $\mathbf{W}^k$ into a product of $\mathbf{W}^{k-p}$ and $\mathbf{W}^p$ with $p \in \mathbb{N}$; step (2) replaces $\alpha_k$ by the convolution $\alpha_k =: \sum_{p=0}^k f_p f_{k-p}$; step (3) changes the summation indices; step (4) arrives at the desired form.

**Convergence.** The brief analysis above has made some implicit assumptions. First, in the definition $\mathbf{M}_\alpha(\mathcal{G}) := \sum_{k=0}^\infty \alpha_k \mathbf{W}^k$, we have taken that the Taylor series converges. This occurs if all the eigenvalues of $\mathbf{W}$ lie within the series' radius of convergence, meaning $\rho(\mathbf{W}) < (\limsup_{n\to\infty} \sqrt[n]{|\alpha_n|})^{-1}$. Here, $\rho(\mathbf{W}) := \max\{|\lambda_1|, ..., |\lambda_N|\}$ denotes the spectral radius. This is a common problem in the graph node kernel literature. It is typically ensured by regularising $\mathbf{W}$ to control its spectral radius (Chung, 1997). We have also taken that $\mathbf{M}_\alpha(\mathcal{G})$ is positive definite, meaning all its eigenvalues are greater than than 0. This can be ensured e.g. by taking $\alpha_0 > 0$, again regularising $\mathbf{W}$ to control its spectral radius, and appealing to Weyl's perturbation inequality (Bai et al., 2000).

We have also assumed that $(f)_{i=0}^\infty$ exists which satisfies the discrete convolution relation, and that the Taylor series $\mathbf{\Phi}_\mathcal{G} = \sum_{i=0}^\infty f_i \mathbf{W}^i$ also converges. Recall that

$$
(1 - z)^{\frac{1}{2}} = \sum_{n=0}^\infty (-1)^n \binom{\frac{1}{2}}{n} z^n,
\tag{14}
$$

which is well-known to converge if $\|z\| \leq 1$. Letting $z = 1 - \mathbf{M}_\alpha(\mathcal{G})$,

$$
\mathbf{M}_\alpha(\mathcal{G})^{\frac{1}{2}} = \sum_{n=0}^\infty (-1)^n \binom{\frac{1}{2}}{n} (1 - \mathbf{M}_\alpha(\mathcal{G}))^n = \sum_{n=0}^\infty (-1)^n \binom{\frac{1}{2}}{n} (1 - \sum_{i=0}^\infty \alpha_i \mathbf{W}^i)^n.
\tag{15}
$$

To converge, we require that $\rho(1 - \mathbf{M}_\alpha(\mathcal{G})) \leq 1$. Since $\mathbf{M}_\alpha(\mathcal{G})$ is positive definite, it is sufficient that $\rho(\mathbf{M}_\alpha(\mathcal{G})) \leq 2$. Again, this is achieved by regularising $\mathbf{W}$ to control its spectral radius and choosing suitable $(\alpha_i)_{i=0}^\infty$. Inspecting Eq. 15, it is clear that $\mathbf{M}_\alpha(\mathcal{G})^{\frac{1}{2}}$ can be expanded in powers of $\mathbf{W}$, so we let

$$
\mathbf{M}_\alpha(\mathcal{G})^{\frac{1}{2}} = \sum_{k=0}^\infty f_k \mathbf{W}^k.
\tag{16}
$$

Substituting this back in, we recover the convolution relationship $\alpha_i =: \sum_{k=0}^i f_k f_{i-k}$. One can find the deconvolution $(f_k)_{k=0}^\infty$ given a sequence $(\alpha_i)_{i=0}^\infty$ using an iterative formula, given in Eq. 6 by Reid et al. (2024b). We take $\mathbf{\Phi}_\mathcal{G} = \sum_{k=0}^\infty f_k \mathbf{W}^k$, then $\mathbf{M}_\alpha(\mathcal{G}) = \mathbf{\Phi}_\mathcal{G} \mathbf{\Phi}_\mathcal{G}^\top$ since $\mathbf{W}$ is symmetric.

**The case for implicit kernel learning.** We have seen that, to guarantee convergence and positive definiteness of $\sum_{k=0}^\infty \alpha_k \mathbf{W}^k$, one must in general control the spectrum of $\mathbf{W}$. This can be achieved by normalising it by suitably downscaling the weights (Chung, 1997; Reid et al., 2024b), or choosing $(\alpha_k)_{k=0}^\infty$ to stay within the radius of convergence. To avoid a tricky constrained optimisation, we therefore prefer to instead directly optimise the set of coefficients $(f_i)_{i=0}^{i_{\max}}$, up to some maximum order $i_{\max}$ above which $f$ vanishes. In doing so, we automatically guarantee both convergence and positive definiteness: the learned $\mathbf{\Phi}_\mathcal{G}$ must be finite and they implicitly defines a kernel in feature space (Gretton, 2013). This is why we choose optimise mask parameters in Alg. 1, elegantly sidestepping any convergence and positive definiteness issues.

## A.2 PROOF OF THM. 3.2 (NOVEL)

The following concentration bounds are new. Referring back to Eq. 10, recall that graph random features are computed by

$$\widehat{\phi}_{\mathcal{G}}(v_i)_q := \frac{1}{n} \sum_{k=1}^{n} \psi(\omega_k^{(i)})_q, \quad q = 1, ..., N, \tag{17}$$

where we define the *projection vector* $\psi(\cdot) : \Omega \to \mathbb{R}^N$ such that

$$\psi(\omega_k^{(i)})_q := \sum_{\omega_{iq} \in \Omega_{iq}} \frac{\widetilde{\omega}(\omega_{iq}) f_{\mathrm{len}(\omega_{iq})}}{p(\omega_{iq})} \mathbb{I}(\omega_{iq} \text{ prefix subwalk of } \omega_k^{(i)}). \tag{18}$$

$\Omega_{iq}$ is the set of walks between nodes $v_i$ and $v_q$; $p(\omega_{iq})$ is the probability of a particular walk $\omega_{iq}$, known from the sampling mechanism; $\widetilde{\omega}(\omega_{iq})$ is a function that returns the product of weights of edges traversed by $\omega_{iq}$ (Eq. 9); $\mathrm{len}(\omega_{iq})$ is the length of $\omega_{iq}$; and $f : \mathbb{N} \to \mathbb{R}$ is a sequence of a real numbers that determines the graph node kernel to be estimated. Following convention, we will assume that we sample random walks with *geometrically distributed lengths*, halting with probability $p_{\mathrm{halt}}$ at each timestep, though in principle different importance sampling schemes are possible.

Suppose we are to estimate a single entry of the graph node kernel, $\mathbf{M}_\alpha(\mathcal{G})_{ij}$. We do so by taking

$$\widehat{\mathbf{M}}_{\alpha ij} := \widehat{\phi}_{\mathcal{G}}(v_i)^\top \widehat{\phi}_{\mathcal{G}}(v_j) = \frac{1}{n^2} \sum_{k_1=1}^{n} \sum_{k_2=1}^{n} \psi(\omega_{k_1}^{(i)})^\top \psi(\omega_{k_2}^{(j)}) \tag{19}$$

where $\mathbb{E}(\widehat{\mathbf{M}}_{\alpha ij}) = \mathbf{M}_{\alpha ij}$ since the estimate is unbiased. We sample $n$ independent random walks starting from every node of the graph.

Let us now consider a *single* pair of random walks from $v_i$ and $v_j$ respectively. Denote this by the random variable

$$X_k := (\omega_k^{(i)}, \omega_k^{(j)}) \in \Omega \times \Omega, \tag{20}$$

where $n$ such random variables $\{X_k\}_{k=1}^{n}$ are used in total. Suppose we modify $X_k$. Under mild assumptions, we can prove that the change in the estimator $|\Delta\widehat{\mathbf{M}}_{\alpha ij}|$ is *bounded*, which will enable us to use derive concentration inequalities.

This can be understood as follows. To begin, we will bound the $L_1$-norm of $\psi(\omega_k^{(i)})$. Observe that

$$\|\psi(\omega_k^{(i)})\|_1 \le \max_{\omega_k^{(i)} \in \Omega} \sum_{\omega \text{ p.s. } \omega_k^{(i)}} \left| \frac{\widetilde{\omega}(\omega) f_{\mathrm{len}(\omega)}}{p(\omega)} \right| \le \sum_{k=0}^{\infty} |f_k| \left( \max_{v_i, v_j \in \mathcal{N}} \frac{|\mathbf{W}_{ij}| d_i}{1 - p_{\mathrm{halt}}} \right)^k =: c, \tag{21}$$

which corresponds to the case of an infinite length walk that happens almost never. Here, $\omega$ p.s. $\omega_k^{(i)}$ means that $\omega$ is a 'prefix subwalk' of $\omega_k^{(i)}$, so the first $\mathrm{len}(\omega)$ nodes of $\omega_k^{(i)}$ are exactly the nodes of $\omega$.

Suppose that $c$ is finite. This can be guaranteed for bounded $f$ by normalisation of $\mathbf{W}$, multiplying by a scalar so that $|\max_{v_i, v_j \in \mathcal{N}} \frac{\mathbf{W}_{ij} d_i}{1-p}| < 1$. More pragmatically, it can be achieved by stipulating that there exists some integer $i_{\max} \in \mathbb{N}$ such that $f_i = 0 \, \forall \, i > i_{\max}$, so we only consider walks up to a certain length. This is a natural choice in experiments since we want to keep the set of learnable parameters finite. It also coincides with the intuition that very long walks should not reflect the relationship between the nodes of a graph of finite size.

We have seen that $0 < \|\psi(\omega_k^{(i))})\|_1 \le c$. It follows that $-c^2 \le \psi(\omega_{k_1}^{(i)})^\top \psi(\omega_{k_2}^{(j)}) \le c^2 \, \forall \, (k_1, k_2, i, j)$, whereupon the change in $\widehat{\mathbf{M}}_{\alpha ij}$ if we modify $X_k$ obeys

$$|\Delta\widehat{\mathbf{M}}_{\alpha ij}| \le 2 \frac{2n-1}{n^2} c^2. \tag{22}$$

With straightforward application of McDiarmid's inequality (Doob, 1940), it follows that

$$\Pr\left( |\widehat{\phi}_{\mathcal{G}}(v_i)^\top \widehat{\phi}_{\mathcal{G}}(v_j) - \mathbf{M}_{\alpha ij}| > t \right) \le 2 \exp\left( -\frac{t^2 n^3}{2(2n-1)^2 c^4} \right), \tag{23}$$

completing the proof. □

As noted in the main text, this is a remarkably general and powerful result. It is the first known concentration inequality for GRFs.

## B  EXTENDED DISCUSSION ON RELATED WORK

In this appendix, we give a detailed explanation of the relationship to previous work.

1. **Stochastic position encoding (SPE)** (Liutkus et al., 2021). To our knowledge, this is the earliest RPE strategy compatible with $\mathcal{O}(N)$ transformers, incorporating information about the 'lag' between tokens in the linear attention setting by drawing samples from random processes with a prescribed covariance structure. The time complexity is $\mathcal{O}(N)$ but the method is only applicable to sequence data, i.e. the very special case that $\mathcal{G}$ is a 1-dimensional grid.

2. **Toeplitz masks and the fast Fourier transform (FFT)** (Luo et al., 2021). Again considering sequence data, suppose that RPE between a query $\boldsymbol{q}_i$ and a key $\boldsymbol{k}_j$ is impemented by adding a bias term $b_{j-i}$ to the dot product before exponentiating. Then the corresponding mask $\mathbf{M}_{ij} = e^{b_{j-i}}$ is *Toeplitz* (constant on all diagonals). From numerical linear algebra, multiplying a Toeplitz matrix and a vector can be achieved with $\mathcal{O}(N \log N)$ time complexity rather than $\mathcal{O}(N^2)$, using the FFT. This is subquadratic, though not linear, and again only specific $\mathcal{G}$ are considered.

3. **Graphormer** (Ying et al., 2021). In this paper, positional encodings based on the shortest path distances between the nodes of the graph (spatial encoding) and their respective degrees (centrality encoding) are seen to substantially improve transformer performance, closing the gap with graph neural networks. There is no attempt to incorporate these techniques with linear attention, but the paper provides strong evidence of the effectiveness of graph-based attention modulation (topological masking).

4. **Block Toeplitz matrices and differential equations on graphs** (Choromanski et al., 2022). This paper proves that subquadratic matrix-vector multiplication is sufficient for masking to be applied efficiently, and suggests a number of algorithms to achieve this in special cases.
   - *Block Toeplitz matrices.* This method extends the work of Luo et al. (2021) to show that the analogous mask on a more general $d$-dimensional grid is $d$-level *block* Toeplitz. Hence, this mask parameterisation also supports $\mathcal{O}(N \log N)$ masking, but now for a slightly more general class of graphs.
   - *Affine mappings on forests.* If $\mathcal{G}$ is a forest and $\mathbf{M}_{ij} = \exp(\tau(\text{dist}(v_i, v_j)))$, where $\text{dist}(v_i, v_j)$ is the shortest path distance between nodes $v_i$ and $v_j$ and $\tau$ is an affine mapping, then topological masking can be implemented in $\mathcal{O}(N)$. The algorithm uses dynamic programming techniques for rooted trees. Notwithstanding its speed, the flexibility of this parameterisation is limited and only particular topologies can be treated.
   - *Heat kernels on hypercubes.* Kondor and Lafferty (2002) derived a closed form for the heat kernel on a hypercube, which also turns out give a Toeplitz mask that can be applied in $\mathcal{O}(N \log N)$. This is a special instantiation of our power series masks $\mathbf{M}_\alpha(\mathbf{W})$ for a specific choice of $\mathcal{G}$. The authors did not empirically test heat kernels; we found them to perform very poorly compared to our more general, faster method.
   - *Random walk graph node kernels.* Here, the authors define an ad-hoc graph kernel by taking the dot product between two 'frequency vectors', recording the nodes visited by the random walks beginning at the two nodes of interest. This method is the closest to our approach, but with some crucial differences: (i) we use *importance* sampling of random walks, which we find to be crucial for good performance (see App. C); (ii) we use *learnable* weights that depend of the length of random walk, permitting flexible upweighting or downweighting of longer- and shorter-range interactions; (iii) our method provides a Monte Carlo estimate of a learnable function of a weighted adjacency matrix, which enjoys certain regularisation properties (Reid et al., 2024b). More importantly, we are able to derive strong concentration bounds that *prove* that our method is $\mathcal{O}(N)$, whereas Choromanski et al. (2022) do not explicitly guarantee any time complexities.

5. **Centrality degree encoding** (Chen et al., 2024). This paper takes the mask $\mathbf{M}_{ij} = \sin(\frac{\pi}{4}[z(d_i) + z(d_j)])$, where $d_{\{i,j\}}$ is the degree of node $v_{\{i,j\}}$ and $z : \mathbb{N} \to (0, 1)$ is a learnable

| | ImageNet 1M | iNaturalist2021 2.7M | Places365 (Small) 1.8M |
|---|---|---|---|
| Num. layers | 12 | 12 | 12 |
| Num. heads | 12 | 12 | 12 |
| Num. patches | $16 \times 16$ | $16 \times 16$ | $16 \times 16$ |
| Hidden size | 768 | 768 | 768 |
| MLP dim. | 3072 | 3072 | 3072 |
| Optimiser | Adam | Adam | Adam |
| Epochs | 90 | 5 | 5 |
| Base learning rate | $3 \times 10^{-3}$ | $3 \times 10^{-3}$ | $3 \times 10^{-3}$ |
| Final learning rate | $1 \times 10^{-5}$ | $1 \times 10^{-5}$ | $1 \times 10^{-5}$ |
| Learning rate schedule | Linear warmup ($10^4$ steps), constant, cosine decay | | |
| Batch size | 4096 | 64 | 64 |
| $\phi(\cdot)$ | ReLU$(\cdot)$ | ReLU$(\cdot)$ | ReLU$(\cdot)$ |
| Pretrained? | ✗ | ✓ | ✓ |
| Num. walks | 100 | 20 | 20 |
| $p_{\text{halt}}$ | 0.1 | 0.1 | 0.1 |
| Max. walk length ($i_{\text{max}}$) | 100 | 10 | 10 |

Table 2: Architecture, hyperparameters and training details for ViT experiments.

function thereof. This can be implemented with linear transformers on general graphs in $\mathcal{O}(N)$ time complexity, but since it is completely insensitive to the relative positions of the nodes in the graph it cannot be considered a general topological masking mechanism.

6. **Kernel decomposition linear graph transformer** (KDLGT; Wu et al.) This paper also attempts to efficiently incorporate topological information into graph transformers by defining a graph node kernel with a low-rank decomposition. It is based off a 'structure extractor' for a subgraph centred at each respective node or the 'shortest anchor path distance'. However, bias is incorporated as an *additive* term, rather multiplicatively which is our chief goal (Eq. 8).

7. **Graph attention networks** (Veličković et al., 2018). This seminal work incorporates masked self-attention into graph neural networks, with each node attending only to its neighbours. The focus is on developing a convolution-style neural network for graph-structured data rather than on efficient masking of linear attention transformers. Our mask parameterisation is more general than this hard nearest-neighbours example, but shares the finding that modulating attention by local topological structure boosts performance.

## C  FURTHER EXPERIMENTS AND DETAILS

Here, we include extra experiments and details too long for the main text.

### C.1  ARCHITECTURES, HYPERPARAMETERS AND TRAINING DETAILS

Table 2 gives details for the ViT experiments.

### C.2  VIT ABLATIONS

Here, we run extra experiments to further evidence the findings reported in the main text. Unless explicitly stated, architectures and training details are as reported in Table 2.

**Number of walkers** $n$. We construct GRFs using $\{1, 10, 100, 1000\}$ walkers with a termination probability $p_{\text{halt}} = 0.5$, retraining from scratch using the respective topological masks. See Fig. 6. Larger $n$ reduces the mask variance and improves predictive performance, but also makes the features $\{\widehat{\phi}_{\mathcal{G}}(v_i)\}_{v_i \in \mathcal{N}}$ less sparse so increases computational cost. Since the estimator is unbiased, the $n \to \infty$ limit gives dense features $\{\phi_{\mathcal{G}}(v_i)\}_{v_i \in \mathcal{N}}$ whose dot product is exactly a Taylor mask $\mathbf{M}_\alpha(\mathcal{G})$.

**Importance sampling.** The crux of our approach is modulating attention by an unbiased Monte Carlo approximation of a function of a weighted adjacency matrix $\mathbf{W}$. Using GRFs, this involves

sampling random walks which we reweight by (i) their probability under the sampling mechanism, (ii) the product of traversed edge weights $\widetilde{\omega}(\omega)$, and (iii) a learnable function $f$ (see Eq. 10). To investigate the importance of this principled sampling mechanism, we ablate (i) and (ii) and compare with an ad-hoc empirical random walk kernel defined by the features

$$\phi_{\text{ad-hoc}}(v_i)_q := \frac{1}{n}\sum_{k=1}^{n}\sum_{\omega_{iq}\in\Omega_{iq}} f_{\text{len}(\omega_{iq})}\mathbb{I}(\omega_{iq} \text{ prefix subwalk of } \omega_k^{(i)}), \quad q = 1, ..., N. \quad (24)$$

Eq. 24 still simulates random walks and records their destinations, giving an empirical measure of the overlap between respective nodes' neighborhoods. However, in absence of dependence on $\widetilde{\omega}(\omega)$ and $p(\omega)$ this cannot be interpreted as a MC estimate of a function of $\mathbf{W}$. This is similar to the approach employed by Choromanski et al. (2022) but with extra learnability via $f$.

Comparing ViT performance with $n = 100$ random walks and $p_{\text{halt}} = 0.5$, the results are striking; see Table 3. In contrast to our method, the mask defined by Eq. 24 is *worse than the unmasked baseline*. This shows the crucial importance of the theory in Sec. 3.4. Intuitively, without upweighting long, improbable walks by $p(\omega)^{-1}$ it is difficult to capture long-range dependencies in the topological mask.

**Fully learnable features.** As a final check, we endow each graph with a fully learnable, dense $N$-dimensional feature $\phi_{\text{dense, learned}}(v_i) \in \mathbb{R}^N$, relaxing the parameterisation as a function of a weighted adjacency matrix $\mathbf{M}_\alpha(\mathcal{G})$. This uses many more parameters and is $\mathcal{O}(N^2)$ rather than $\mathcal{O}(N)$, but it technically includes GRFs as a special case. In making this choice we have removed all structural inductive bias from $\mathcal{G}$, allowing the model to implicitly learn a masking kernel without any constraints (except feature dimension). The model performs badly, providing only a very slight improvement over the unmasked linear baseline. See Table 3.

Figure 6: Final test accuracy vs. number of walkers $n$. Monte Carlo mask variance drops with the number of walks so predictions improve.

Table 3: Ablation studies. Importance sampling of random walks and mask parameterisation as a function of $\mathbf{W}$ are crucial for good performance.

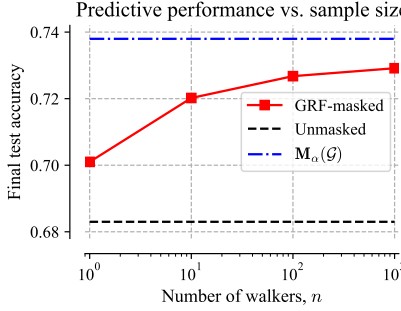

| Variant | Test acc. |
|---|---|
| **GRF-masked** | 0.730 |
| Unmasked | 0.693 |
| Ad-hoc masked (Eq. 24) | 0.689 |
| Fully learnable features | 0.696 |

**Time complexity scaling with number of walkers.** Readers may also be interested in how the total number of FLOPs scales with the number of walkers $n$, for a given graph $\mathcal{G}$ and termination probability $p_{\text{halt}}$. There are two main contributions to consider. First, one must *sample* the random walks, the cost of which is $\mathcal{O}(n)$. This is done once at the start of training, so is not a big consideration in practice. Second, the number of walks determines the sparsity of the GRF, and therefore the time complexity of implicit masking via Eq. 7. We have see that this is at most $\mathcal{O}(n)$, but the details will vary depending on the graph topology (see Lemma 3.3). Fig. 7 gives an example for a 1D grid with $n = 64$, taking $p_{\text{halt}} = 0.5$ and $d = m = 8$. It reflects the behaviour described above.

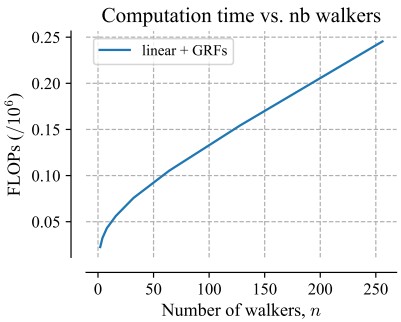

Figure 7: Time complexity scaling wrt number of walkers $n$.

**Linear attention variant.** As a brief note, our topological masking algorithm is compatible with different choices for the nonlinearity $\phi(\cdot)$. In the text we have generally taken $\phi(\cdot) = \text{ReLU}(\cdot)$ so that $m = d$, but other choices are possible. For instance, one could use a randomised mapping that approximates the softmax kernel (Choromanski et al., 2020; Likhosherstov et al., 2024), which is commonly referred to as the *Performer*.

To demonstrate, we replace the ReLU features by FAVOR+ Monte Carlo attention approximation, considering the same ViT architecture and training as reported in Sec. 4 (see Table 2). For ImageNet, the unmasked Performer achieves a test accuracy of **0.693**. Meanwhile, the GRF-masked performer achieves a test accuracy of **0.734** – a 3.9% absolute improvement. This shows how Alg. 1 can be expected to provide gains across a range of linear attention mechanisms.

### C.3  HD-VPD EXPERIMENTAL DETAILS

Here, we describe the high density visual particle dynamics (HD-VPD) experiment from Sec. 4 in more detail. The setup closely follows that of (Whitney et al., 2024), to which we direct the interested reader for an exhaustive information about hardware and modelling techniques. We will provide a brief overview for convenience.

**Dynamics modelling as video prediction.** The HD-VPD stack involves three steps: (1) encode the input images as 3D point clouds; (2) predict updates to the point clouds, conditioned on robot actions; (3) render the predicted new state into an image. We train end-to-end using video prediction loss. In more detail:

1. *Encode the input images.* The model receives two RGB-D (colour channels plus depth) images at times $t - 1, t - 2$. Each RGB image is encoded into per-pixel features using a U-Net (Ronneberger et al., 2015), and unprojected using known camera intrinsics and extrinsics to a point cloud $(\boldsymbol{x}_i, \boldsymbol{f}_i)_{i=1}^{N_{\text{pixels}}}$ with $\boldsymbol{x} \in \mathbb{R}^3$ and $\boldsymbol{f} \in \mathbb{R}^{16}$. For every timestep, the particles across different cameras are merged and subsampled uniformly at random to get $N = 32768$. The robot actions at timesteps $\{t - 2, t - 1, t\}$, representing the robot joints and where they plan to move, are also encoded as *kinematic particles*.

2. *Predict the point cloud dynamics.* With an *Interlacer* (see below), use the point clouds and kinematic particles to predict $(\Delta \boldsymbol{x}_i, \Delta \boldsymbol{f}_i)_{i=1}^N$, and hence the updated state $(\boldsymbol{x}_i^{t-1} + \Delta \boldsymbol{x}_i, \boldsymbol{f}_i^{t-1} + \Delta \boldsymbol{f}_i)_{i=1}^N$.

3. *Render the new state to an image.* Use a ray-based renderer similar to Point-NeRF (Xu et al., 2022) to render an image for the updated point cloud.

The dynamics model can be recursively applied to make predictions multiple rollout timesteps into the future. To train, we sample a subset of rays to render at each timestep and compute the $L_2$ loss between the predicted and observed RGB values.

**Interlacers as PCTs.** The Interlacer is a type of point cloud transformer (PCT) designed to scale to tens of thousands of particles. It alternates linear-attention transformer layers that capture global dependencies between tokens with local *neighbour-attender* layers that model fine-grained geometric structure (see Fig. 3 by Whitney et al. (2024)). This is reported to outcompete both graph neural

networks and vanilla linear-attention transformers. For the neighbour-attender, the authors use a message passing-type algorithm that updates the features of a subset of anchor particles depending on the $k$-nearest neighbours' position and feature vectors, and then updates the remaining particles depending on their closest anchor. In Sec. 4, we show that this can be replaced by GRF-masked linear attention. This also captures local structure and obviates expensive message passing.

**Training and architecture details.** This exactly follows App. E by Whitney et al. (2024), apart from:

1. We average over 5 seeds, discarding one where training does not converge.

2. For our algorithm, we use *asymmetric* GRFs. (Sec. D). We use $n = 3^3 = 27$ repelling random walks (Reid et al., 2023b). Instead of terminating, we sample 3 hops deterministically since the computational bottleneck is finding the $k$-nearest neighbours list rather than sampling lengths once it has been computed. We take $\phi(\cdot) = \text{ReLU}(\cdot)$ and compute GRF-masked attention with a single head, projecting the queries and keys with a dense layer.

3. To construct $\mathcal{G}$, we find $k = 3$ neighbours for every node. This is done on the fly since the neighbours can change at every timestep.

4. For Fig. 4, we train for 100k steps, using a learning rate of $3 \times 10^{-4}$ until step 1000 then $1 \times 10^{-4}$ until step 100k.

To enumerate some other important details: all models are trained with a batch size of 16; we use the AdamW optimiser (Loshchilov, 2017) with weight decay $10^{-3}$, clipping the gradient norm to 0.01; models are trained with 6 step rollouts, with losses computed on 128 sampled rays; $\Delta \boldsymbol{x}_i$ are constrained to $[-0.25, 0.25]$ by a scaled $\tanh$ nonlinearity so particles do not move more than 25cm per timestep. For the rendering, like Whitney et al. (2024) we use four concentric annular kernels of radii $[0, 0.01, 0.02, 0.05]$ with bandwidths $[0.01, 0.01, 0.01, 0.05]$, approximated using 16 nearest neighbours. The near plane is 0.1m and the far plane is 2m, with the background set to solid white.

### C.4   VIVIT: TOPOLOGICAL MASKING FOR VIDEO DATA

Here, we present preliminary results for incorporating topological masking with graph random features into video vision transformers (ViViT; Arnab et al., 2021). Prompted by the original paper, we use a factorised encoder, where the spatial encoder follows the ViT architecture from earlier in the paper (see Sec. 4 and App. C.1) with patch size $16 \times 16 \times 2$. The final index mixes pairs of successive frames. The temporal encoder, which subsequently models interactions between the frame-level representations, also follows this architecture but with 4 layers instead of 12. We train for 30 epochs

Table 4: ViViT test accuracies on the Kinetics benchmark. Topological masking with GRFs boosts performance.

| VARIANT | TEST ACC. |
|---|---|
| SOFTMAX | 0.754 |
| SOFTMAX + GRFs | 0.758 |

using the Adam optimiser, with base learning rate $10^{-1}$ and final learning rate $10^{-4}$. We take the same schedule as in Table 2. We train and evaluate on the Kinetics 400 benchmark (Kay et al., 2017). Table 4 shows the results for (i) unmasked softmax attention and (ii) softmax attention with GRFs. Topological masking boosts predictive performance by $+\mathbf{0.4}\%$. To our knowledge, this is the first application of topological masking to the video modality.

## D   ASYMMETRIC GRAPH RANDOM FEATURES

In this Appendix, we present a variant of our core topological masking algorithm that relaxes the assumption that the function $f$ for the query and key ensembles of walkers are identical, instead using *asymmetric* graph random features. These are believed (but not proved) to generally give higher variance mask estimates (Reid et al., 2024b), but making a judicious choice can also bring computational and implementation benefits. Using asymmetric GRFs for topological masking is also a novel contribution of this paper. In Sec. 4, it is found to be a convenient choice for the Interlacer (Whitney et al., 2023) because it can be straightforwardly swapped into existing code.

**Asymmetric GRFs.** It is straightforward to see that Eq. 13 of App. A.1 generalises to

$$\mathbf{M}_\alpha(\mathcal{G})_{ij} = \sum_{v=1}^{N} \left( \sum_{k_1=0}^{\infty} f_{k_1}^{(1)} \mathbf{W}_{iv}^{k_1} \right) \left( \sum_{k_2=0}^{\infty} f_{k_2}^{(2)} \mathbf{W}_{jv}^{k_2} \right) \tag{25}$$

if the discrete convolution

$$\sum_{p=0}^{k} f_p^{(1)} f_{k-p}^{(2)} = \alpha_k \tag{26}$$

holds. We do not necessarily require that $f^{(1)} = f^{(2)}$; this is a special case. Another special case is $f_p^{(2)} = \{1$ if $p = 0$, $0$ otherwise$\}$, whereupon we need $f_p^{(1)} = \alpha_p \,\forall\, p$. Sampling GRFs in this manner will still give an unbiased estimate of $\mathbf{M}_\alpha(\mathcal{G})_{ij}$, but with different concentration properties.

**Computational benefits.** Supposing that $f_p^{(2)} = \{1$ if $p = 0$, $0$ otherwise$\}$, $\phi_{\mathcal{G}}^{(2)}(v_i)_n = \mathbb{I}(i = n)$, a one-hot vector only nonzero at the coordinate corresponding to $v_i$. In other words, we do not need to simulate any random walks to obtain GRFs for the keys; $\mathbf{\Phi}_{\mathcal{G}}^{(2)} = \mathbf{I}_N$ with $\mathbf{I}_N \in \mathbb{R}^{N \times N}$ the identity matrix. It follows that

$$\widehat{\mathbf{M}}_\alpha(\mathcal{G}) = \mathbf{\Phi}_{\mathcal{G}}^{(1)} \mathbf{\Phi}_{\mathcal{G}}^{(2)^\top} = \mathbf{\Phi}_{\mathcal{G}}^{(1)} =: [\phi_{\mathcal{G}}^{(1)}(v_i)]_{i=1}^{N}. \tag{27}$$

But $\phi^{(1)}(v_i)$ is only nonzero at nodes visited by the ensemble of walkers beginning at $v_i$. This makes masked attention straightforward to efficiently compute, obviating the outer products and vectorisation. Simply,

$$(\mathbf{A} \odot \widehat{\mathbf{M}}_\alpha(\mathcal{G}))\mathbf{V} = (\mathbf{A} \odot \mathbf{\Phi}_{\mathcal{G}}^{(1)})\mathbf{V} = \left[ \sum_{j=1}^{N} \mathbf{A}_{ij} \phi_{\mathcal{G}}^{(1)}(v_i)_j \boldsymbol{v}_j \right]_{i=1}^{N} \tag{28}$$

where $\mathbf{A}_{ij} = \exp(\boldsymbol{q}_i^\top \boldsymbol{k}_j)$ (softmax) or $\mathbf{A}_{ij} = \boldsymbol{q}_i^\top \boldsymbol{k}_j$ (linear) and $\{\boldsymbol{v}_j\}_{j=1}^{N}$ are the value vectors. Since $\phi_{\mathcal{G}}^{(1)}(v_i)$ is sparse, the sum in Eq. 28 is computed in constant time so attention is computed in $\mathcal{O}(N)$. Even more explicitly, for an ensemble of walks $\{\omega_k^{(i)}\}_{v_i \in \mathcal{N}, \, k \in [\![1,n]\!]}$, we can rewrite the above as

$$\left[ \sum_{k=1}^{n} \sum_{\omega \text{ p.s. } \omega_k^{(i)}} \mathbf{A}_{i\omega[-1]} \frac{\widetilde{\omega}(\omega) f_{\text{len}(\omega)}}{p(\omega)} \boldsymbol{v}_{\omega[-1]} \right]_{i=1}^{N} \tag{29}$$

where $\omega[-1]$ stands for the final node of the walk $\omega$ and p.s. means 'prefix subwalk' (see App. A.2). The number of FLOPs to compute attention for node $v_i$ depends on the length of all the geometrically distributed walks beginning at node $v_i$, but we have seen that this is independent of graph size (Sec. 3.4). Computing the entire matrix is only $\mathcal{O}(N)$. We give pseudocode for this approach in Alg. 2.

**Benefits and limitations of asymmetric GRFs.** The chief benefit of asymmetric GRFs is that Eq. 29 makes efficient masked attention very straightforward to compute, without relying on sparse matrix operations that may not be well-optimised in machine learning libraries. We use it for the HD-VPD experiments because it can be easily integrated with the existing code: instead of finding each node's (approximate) $k$-nearest neighbours and implementing the complicated message passing-type algorithm by Whitney et al. (2024), we find the 3-hop neighbourhood and straightforwardly compute weighted attention with a learnable function of walk length. The drawback is that the variance of the mask estimate with asymmetric GRFs tends to be greater, though proving this is an open problem.

## E    ADDENDUM: ADDITIONAL GRF RESULTS

Thm. 3.2 provides a novel bound on the GRF estimator concentration, specifying the number of walkers $n$ and termination probability $p_{\text{halt}}$ needed to ensure sharp mask estimates with high probability. Under mild assumptions on graph (namely, that $c$ does not change as $N$ grows), this can be used to upper bound GRF sparsity and thus the time complexity of the algorithm. The bound derived in Lemma 3.3 implicitly assumes the worst case that walkers never backtrack and do not visit

---

**Algorithm 2** $\mathcal{O}(N)$ topological masking with *asymmetric* GRFs

---

**Input:** query matrix $\mathbf{Q} \in \mathbb{R}^{N \times d}$, key matrix $\mathbf{K} \in \mathbb{R}^{N \times d}$, value matrix $\mathbf{V} \in \mathbb{R}^{N \times d}$, graph $\mathcal{G}$ with weighted adjacency matrix $\mathbf{W} \in \mathbb{R}^{N \times N}$, learnable mask parameters $\{f_i\}_{i=0}^{i_{\max}}$, number of random walks to sample $n \in \mathbb{N}$, query/key feature map $\phi(\cdot) : \mathbb{R}^d \to \mathbb{R}^m$.

**Output:** masked attention $\mathrm{Att}_{\mathrm{LR},\widehat{\mathbf{M}}}(\mathbf{Q}, \mathbf{K}, \mathbf{V}, \mathcal{G})$ in $\mathcal{O}(N)$ time, without using sparse linear algebra libraries.

1: Simulate $n$ terminating random walks $\{\omega_k^{(i)}\}_{k=1}^n$ out of every node $v_i \in \mathcal{N}$
2: **for** $v_i \in \mathcal{N}$ **do**
3:     Initialise output feature, $\boldsymbol{f}_i \leftarrow \mathbf{0}_d$
4:     Initialise attention normalisation, $S_i \leftarrow 0$
5:     **for** $\omega^{(i)}$ in $\{\omega_k^{(i)}\}_{k=1}^n$ **do**
6:         **for** $(t, v_j)$ in enumerate($\omega^{(i)}$) **do**
7:             $\boldsymbol{f}_i \mathrel{+}= \phi(\boldsymbol{q}_i)^\top \phi(\boldsymbol{k}_j) \times f_t \times \frac{\widetilde{\omega}(\omega^{(i)}[:t])}{p(\omega^{(i)}[:t])} \times \boldsymbol{v}_j$
8:             $S_i \mathrel{+}= \phi(\boldsymbol{q}_i)^\top \phi(\boldsymbol{k}_j) \times f_t \times \frac{\widetilde{\omega}(\omega^{(i)}[:t])}{p(\omega^{(i)}[:t])}$
9:         **end for**
10:     **end for**
11:     Normalise output feature, $\boldsymbol{f}_i / = (S_i \times n)$
12: **end for**
13: Return low-rank attention with topological masking in $\mathcal{O}(N)$ time by

$$\mathrm{Att}_{\mathrm{LR},\widehat{\mathbf{M}}}(\mathbf{Q}, \mathbf{K}, \mathbf{V}, \mathcal{G}) = [\boldsymbol{f}_i]_{i=1}^N \qquad (30)$$

---

any of the same nodes, so in practice how *tight* this bound is may depend on the particular graph being considered. In particular, it will tend to be tighter for denser, bigger graphs. Our algorithm is even more efficient for smaller, sparser graphs. This behaviour, beyond asymptotic time complexity results, may be of interest to practitioners.

To supplement with some empirical results, we consider using GRFs to estimate the heat kernel $\exp(\sigma^2 \mathbf{W})$, where $\sigma \in \mathbb{R}$ is a lengthscale parameter and $\mathbf{W}_{ij} = 1/\sqrt{d_i d_j}$ as usual. We consider Erdős-Rényi graphs, where edges are independent Bernoulli random variables with probability $p$.

**Scaling with graph sparsity.** The left pane of Fig. 8 shows the Frobenius norm error on the mask estimate, $\|\widehat{\mathbf{M}} - \mathbf{M}\|_2 / \|\mathbf{M}\|_2$ where $\widehat{\mathbf{M}}$ is the estimate, vs. the edge generation probability $p$. The number of nodes is $N = 100$, and we take $n = 4$ walks per node with $\sigma = 0.5$. We average the results over 100 trials for standard errors. As expected, the approximation error becomes a little worse for denser graphs, though it remains excellent throughout. The right hand pane of Fig. 8 shows the RF sparsity (proportion of nonzero entries in $\widehat{\mathbf{\Phi}}_{\{\mathbf{Q},\mathbf{K}\},\mathcal{G}}$) over the same parameters. Again, the number of nonzero entries grows as graphs become denser because visiting the same node multiple times is less likely, but GRFs are sparse in every case. *We emphasise once more that the bounds still hold in every case; this is simply a matter of how tight they are.*

**Scaling with graph size.** Fig. 9 performs the same analysis for Erdős-Rényi graphs with fixed edge probability $p = 0.5$, now varying the size of the graph $N$. The approximation error grows slightly for bigger graphs but soon plateaus, and again remains excellent for all graphs considered. The GRF sparsity drops because the number of visited nodes is $\mathcal{O}(1)$ in every instance, meaning the *proportion* on nonzero GRF entries must scale as $\mathcal{O}(1/N)$.

We have included this section for the convenience of the interested reader. For more exhaustive details, we invite them to read the previous works of Reid et al. (2024b) and Choromanski (2023), which first introduced GRFs and explored their properties.

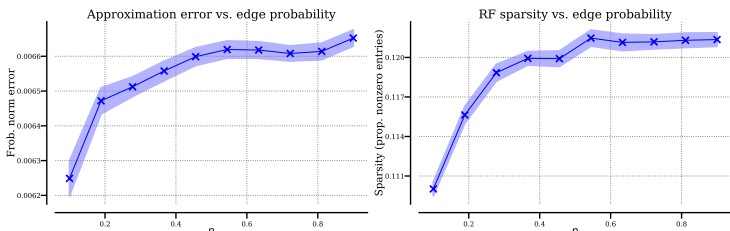

Figure 8: GRF approximation quality and sparsity for Erdős-Rényi graphs of different sparsities.

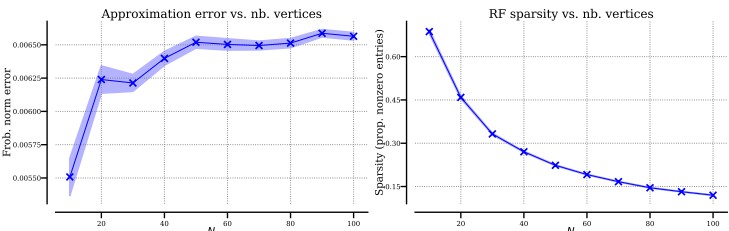

Figure 9: GRF approximation quality and sparsity for Erdős-Rényi graphs of different sizes.

