# OpenReview forum: "Linear Transformer Topological Masking with Graph Random Features"
_ICLR.cc/2025/Conference — ICLR 2025 Poster_

### Official Review · Reviewer_uZmX · 2024-10-20

**Soundness:** 3
**Presentation:** 3
**Contribution:** 2
**Rating:** 6
**Confidence:** 2

**Summary:**

This paper presents a novel approach by parameterizing topological masks as a learnable function of a weighted adjacency matrix. This method incorporates a strong structural inductive bias with rigorous concentration bounds, improving both time and space complexity.

**Strengths:**

1. The research topic is highly important and intriguing and the authors provide experimental results on some larger scale image and point-cloud datasets.
2. Generally, the notation and theorem in the paper is well defined and illustrated.
3. The authors provide the theoretical analysis and ablation study to show the effectiveness of the proposed method.

**Weaknesses:**

#### Major problems
1. The background and related work section is limited. To my knowledge, this is not the first work to incorporate the inductive bias inherent in graph nodes into GNN and transformer-like models.
2. Additionally, improved time and space complexity have been discussed in existing papers such as [A, B]. Therefore, the benefits and differences between linear attention and the proposed topological masking should be thoroughly explained.
3. It's better to include more experiments and evaluations on traditional graph datasets.

[A] Wu, Qitian, et al. "Nodeformer: A scalable graph structure learning transformer for node classification." NIPS2022.
[B] Wu, Yi, et al. "KDLGT: A Linear Graph Transformer Framework via Kernel Decomposition Approach." IJCAI 2023.
#### Minor issues
1. Confusing notations in line 148, the size of the graph N ($\mathcal{N}$)?

**Questions:**

See the weakness.

---

> ### Author Response · Authors · 2024-11-14
> **Thanks for the review**
>
> We thank the reviewer for their comments. We are pleased that they think our research is important, and that they appreciate the theoretical contributions and ablation analysis. We respond to all their questions and concerns below.
>
> 1. *`Background and related works section is limited’*. **We respectfully point the reviewer to Appendix B, which gives a very detailed account of the relationship to existing work**. We wonder whether they may have missed it, and will be sure to signpost it better in the main body. With regards to the specific papers they suggest:
> - Nodeformer [1]: This paper focuses on *learning* latent graph structures, rather than incorporating information about an existing graph information as a structural inductive bias. It is not directly relevant.
> - KDLGT [2]: This paper does indeed incorporate a structural inductive bias into Transformers, but it does so with a simple additive RPE mechanism. This is not able to modulate the attention by a multiplicative graph function, which is our core goal (see Eq. 2). Nonetheless, we agree that it might be of interest to readers so have added a brief description in the Appendix. Thanks for the pointer.
> 2. *Experiments and evaluations on traditional graph datasets*. Respectfully, we already include experiments on *three* different data modalities: images, videos and point clouds for robotics. We also include extensive ablation studies. Experiments like node classification on datasets such as Citeseer are not the focus of this paper, but might make for interesting future work in a separate publication.
> 3. *Confusing notation in line 148*. $\mathcal{N} = \\{v_1, v_2, …, v_N \\}$ denotes the set of graph nodes. $N$ is the size of this set, i.e. the total number of graph nodes. In the Transformer context, this is equal to the number of tokens. This notation is very standard.
>
> Once again, we thank the reviewer. We have pointed them to our existing related work section (now supplemented with their additional suggestion), and clarified some points of minor misunderstanding. We very much hope that, on reflection, they will raise their score.
>
> _____________________
> [1] Nodeformer: A Scalable Graph Structure Learning Transformer for Node Classification, Wu et al., NeurIPS 2022, https://arxiv.org/abs/2306.08385
> [2] KDLGT: A Linear Graph Transformer Framework via Kernel Decomposition Approach, Wu et al., IJCAI 2023, https://www.ijcai.org/proceedings/2023/0263.pdf

---

> > ### Comment · Reviewer_uZmX · 2024-11-16
> >
> > I have read author's rebuttal, especially on background and related works. I think they have already addressed my main concerns.

---

> > > ### Author Response · Authors · 2024-11-16
> > > **response**
> > >
> > > We would like to sincerely thank the Reviewer uZmX for the comment and ask whether the score can be updated accordingly.
> > >
> > > Yours sincerely,
> > >
> > > The Authors

---

> > > > ### Author Response · Authors · 2024-11-24
> > > > **Any further questions?**
> > > >
> > > > As the discussion period draws to a close, this is a polite reminder that we will be happy to answer any further questions the reviewer may have. We are happy that they say we have addressed all their main concerns. **Respectfully, might they please consider raising their score to reflect this?** Once again, we thank them for their time and efforts.

---

### Official Review · Reviewer_QFRp · 2024-10-31

**Soundness:** 3
**Presentation:** 4
**Contribution:** 3
**Rating:** 8
**Confidence:** 3

**Summary:**

The paper proposes to use the gram matrix of the graph node kernels as the masking matrix for the attention mechanism of Transformer. The fact that each element of the gram matrix can be written as the inner product of two feature vectors (kernel def), we can write the **attention mechanism with masking**  in the form of Equation 3 of the low-ranking setting by redefining $\Phi_Q$ and $\Phi_K$. graph random features are also further applied to reduce the complexity. The experiments are conducted on ViTs and the prediction of the particle dynamics for robotics.

**Strengths:**

* The paper is well-organized and well-written, it's a pleasure to read.
* The reasoning of the idea is clear. We can understand very well why we need to use kernel gram matrix as the masking matrix and why we need the graph random features.
* The experiments are well-presented, which show the relevance of the proposition for the situation where $N$ is large.

**Weaknesses:**

Please see the questions raised.

**Questions:**

* While the true time complexity of the proposed algo is $\mathcal{O}(Nmd)$, most part of the paper omit $m$ and $d$. I think the authors should make this point clear since in some cases where $m$ and $d$ can be large enough.
* The citation in line 161 (Borgwardt et al., 2005) is about graph kernels, not graph node kernels if I understand it well.
* Though the paper shows experimentally the impact of the number of random walks $n$ on the performance, I would also like to see its impact on the computation time.
* In section 4, maybe it's better to use \textit{subsection} for each experiment instead of \textit{paragraph}.
* This question is not about the contribution of the paper, but the general idea of using structural graph masking for transformer. Isn't it a reinvention of the graph neural network by integrating a matrix representing the structure information into Transformer? Can author make a (possible) link between the two?

---

> ### Author Response · Authors · 2024-11-14
> **Thanks for the review**
>
> We thank the reviewer for their comments. We are pleased that they find the paper well organised and written, and that they appreciate the importance of our methods when $N$ becomes large.
>
> 1. *Time complexity is $\mathcal{O}(Nmd)$*. The reviewer is correct that, as with any low-rank attention method with $N$ tokens of dimensionality $d$ and features of dimensionality $m$, the time complexity of Eq. 3 is $\mathcal{O}(Nmd)$. We state this in line 121. Since we are primarily interested in scaling with respect to the number of tokens, we follow convention in the literature by shortening this to $\mathcal{O}(N)$ throughout the text [1,2,3]. For example, for the Interlacer experiment with massive point clouds, $d=64$, $m=16$ but $N=32768$, so the scaling with respect to $N$ is most interesting and important. We believe our notation choice to be standard and unambiguous, and it reduces clutter. However, we agree that it is important to be as clear as possible so will add this comment to the manuscript. Thanks.
> 2. *’Line 161 is about graph kernels, not graph node kernels’*. Thanks for the comment. Graph kernels and graph node kernels are closely related: graph kernels define a product graph using the two constituent graphs, compute the corresponding graph node kernel, then sum up its (weighted) entries. See e.g. page 4 of Bogwardt [4]. For this reason, we still believe the reference to be relevant for our work.
> 3. *Number of walks and computation time*. Thanks for the suggestion. We already include extensive ablations: number of walkers $n$ vs. Transformer performance, the presence or absence of importance sampling, and use of fully learnable features instead of GRFs. Nonetheless, we agree that including number of walkers vs. number of FLOPs would be a nice addition, **so have now added a further plot to Appendix C.2**. The cost of sampling random walks is linear in the number of walkers. The cost of computing the corresponding masked attention in Eq. 7 initially grows at most linearly (by Lemma 3.2), but will eventually saturate once the features become dense. Our plot reflects this. Thanks again for prompting a nice addition.
> 4. *Section 4 headers*. Thanks for the stylistic comments about using section headers instead of paragraphs. We agree that this may be clearer, so have updated the manuscript.
> 5. *Relationship to GNNs*. Thanks for the comment. **We actually already discuss the relationship to GNNs in the paragraph beginning on line 367**. GATs [5] are a particular type of Transformer/GNN with a very strong structural inductive bias, where nodes only attend to their neighbors. Our scheme could be considered a stochastic relaxation of this, still injecting information about the graph but now including longer-range attention by importance sampling random walks. Also, our method can be considered more expressive than GNNs because it is able to distinguish graphs identical under the  1-dimensional Weisfeiler Lehman graph isomorphism heuristic.
>
> We again thank the reviewer for their feedback. We warmly invite them to respond with any further questions, and respectfully ask that they consider raising their score.
>
> ____________
> [1] Rethinking Attention with Performers, Choromanski et al., ICLR 2021, https://arxiv.org/abs/2009.14794
> [2]  From Block-Toeplitz Matrices to Differential Equations on Graphs: Towards a General Theory for Scalable Masked Transformers, Choromanski et al., ICML 2022,  https://doi.org/10.48550/arXiv.2107.07999
> [3] Reformer: The Efficient Transformer, Kitaev et al., ICLR 2020, https://arxiv.org/pdf/2001.04451
> [4] Protein Function Prediction via Graph Kernels, Bogwardt et al., Bioinformatics 2005, https://doi.org/10.1093/bioinformatics/bti1007
> [5] Graph Attention Networks, Veličković et al., ICLR 2018, https://doi.org/10.48550/arXiv.1710.10903

---

### Official Review · Reviewer_w6hQ · 2024-11-01

**Soundness:** 3
**Presentation:** 3
**Contribution:** 3
**Rating:** 8
**Confidence:** 2

**Summary:**

This paper proposes a topological masking method when training transformers on graph-structured data. By decomposing and approximating the graph mask with graph random features, the proposed method achieves linear time and space complexity w.r.t input size. The author shows that their masking algorithm is efficient and high-performance using experiment results.

**Strengths:**

1. The paper has a good motivation for introducing linear topological masking of low-rank attention.

2. The author explains well from introducing the topological mask, using the graph feature to achieve low-rank attention, and leveraging GRF to approximate the graph feature.

3. The explanation is clear, the figures are illustrative, and the writing is well-structured.

**Weaknesses:**

1. While the author emphasizes a lot about the efficiency of the proposed method, the evaluation and experimental parts mainly show the accuracy achieved and lack the corresponding efficiency results like time and memory.

**Questions:**

1. While the author shows the test accuracies in Table 1, can the author also present other results, like the total training time or total flops, to validate the proposed method’s efficiency?

2. In Figure 5, it seems the accuracy improvement achieved by GRF Interlacer is on the starting timestep 0. After several timesteps, it’s becoming similar to MP Interlaced. Can the author explain the reason behind the accuracy improvement at the beginning and the drop?

3. Besides GRF, are there other methods to do implicit graph masking in equation (8)? How's the performance?

If all my concerns are resolved properly, I will be happy to increase my score.

---

> ### Author Response · Authors · 2024-11-14
> **Thanks for the review**
>
> We thank the reviewer for their comments. We are pleased that they agree the paper is well-motivated and that they find the explanations clear. We address all their concerns and questions below.
>
> 1. *‘Can the author also present other results, like the total training time or total flops, to validate the proposed method’s efficiency?’* Thanks for the suggestion. **We actually already include an experiment showing how the total number of FLOPs scales with the input size: see Fig. 3**. As expected, our method is linear. There is a constant multiplicative cost incurred by topological masking with GRFs, and we are much more efficient than the softmax baseline. To supplement with some wall clock times, for the robotics PCT experiment the MP Interlacer baseline [1] (previous SOTA) trains at 0.925 steps/second. Our method, the GRFs Interlacer, trains at 0.982 steps/second. Hence, our method is **not only more accurate, but also faster by 6%**. (The unmasked baseline, which struggles to capture local structure so gives poor accuracy, trains at 1.49 steps/second). We have added these results to the manuscript; thanks for the suggestion. Finally, we emphasise that, for point clouds with $>30k$ nodes, computing full-rank softmax attention is *not even possible* on any reasonable hardware – the time and memory requirements are too great. The fact that we can train our model and make predictions on such a massive graph provides further experimental evidence that it is extremely efficient. **As such, we respectfully suggest that the paper includes plenty of experimental results showing our algorithm’s efficiency, in addition to our detailed theoretical arguments (Sec. 3.4)**.
> 2. *Difference between GRF Interlacer and MP Interlacer*. The GRF Interlacer (our method) uses GRF-masked linear attention, whereas the MP Interlacer [1] uses GNN-style message passing layers. Our method achieves higher image SSIM (prediction accuracy) because it is more expressive and better able to model complex dynamics in the point cloud. The reviewer is correct to note that the difference between the GRF and MP Interlacers seems to become smaller after many rollout timesteps. This may be because applying several GNN layers in sequence gradually increases the receptive field, becoming closer to topological masking (where instead nodes can attend if their ensembles of random walks overlap). However, as is very often the case with deep learning methods, this interpretation is speculative.
> 3. *Besides GRFs, are there any other methods to do implicit graph masking?* A key aspect of our work is that it is **the first linear algorithm for $\mathcal{O}(N)$ topological masking on general graphs** – so no, there is no equally efficient alternative to directly compare against. The closest previous algorithm is block Toeplitz masking [2], which is $\mathcal{O}(N\log N)$ rather than $\mathcal{O}(N)$ and can only be applied to grid graphs. We include this benchmark for the image experiments in Table 1. We find that, despite our method being cheaper, it still tends to match or beat block Toeplitz.
>
> Once again, we thank the reviewer. We believe that we have resolved all their questions and concerns, and have added the extra wall clock times for the Interlacer experiment to the manuscript. We warmly invite them to respond with any further questions, and respectfully request that they consider raising their score.
>
> ________________
> [1] Modelling the Real World with High Density Visual Particle Dynamics, Whitney et al., CoRL 2024, https://arxiv.org/abs/2406.19800
> [2]  From Block-Toeplitz Matrices to Differential Equations on Graphs: Towards a General Theory for Scalable Masked Transformers, Choromanski et al., ICML 2022,  https://doi.org/10.48550/arXiv.2107.07999

---

> > ### Comment · Reviewer_w6hQ · 2024-11-18
> >
> > Thanks for the author addressing my concerns. Now, the contribution statements are better validated through the added results. I have increased my score from 6 to 8. But I would like to keep my confidence score as I am not an expert in this area and give the decision to other reviewers and AC.

---

### Official Review · Reviewer_BeTo · 2024-11-03

**Soundness:** 3
**Presentation:** 1
**Contribution:** 3
**Rating:** 6
**Confidence:** 3

**Summary:**

This paper presents a method for integrating topological graph information into graph transformers through a learnable topological-masking mechanism, using graph random features (GRFs). The authors propose to approximate topological masks via Monte Carlo estimation via GRFs to represent structural biases while ensuring linear-time computation.

**Strengths:**

1. Their method is the first to achieve $\mathcal{O}(N)$-time complexity for computing masked attention for general graphs, $N$ being the number of vertices.
2. The paper provides the first known concentration bounds for GRFs and rigorous sparsity guarantees. These theoretical insights are valuable, potentially extending beyond transformers to other domains that rely on scalable graph-based representations.
3. Their method demonstrates improved predictive performance in various learning tasks.

**Weaknesses:**

1. Dense and unclear presentation:
   - While the method is theoretically sound, the presentation is mathematically dense and lacks clear explanations. This may pose a barrier to readers, particularly those less familiar with GRFs. In particular, the technical exposition in lines 184–254 is notation-heavy and unclear.
   - Algorithmic descriptions, such as those in Algorithm 1, are highly abstract and may be difficult to follow.

   Without clearer explanations, the accessibility of the paper is reduced.
2. The paper lacks discussion of the method limitations. For example:
   - The practical applicability of this method depends heavily on the specifics of the graph structure and the task requirements, since it relies on approximations with random walks. In  graphs where relevant information is distributed over long distances or requires traversing multiple nodes, random walks may fail to capture the full structure efficiently.
   - For dynamic or evolving graphs, precomputing random walks is not feasible, and recomputing them on the fly could reduce efficiency.
   - Since random walks introduce stochasticity, their effectiveness can vary based on the number of walks and the chosen halting probability. This means that the quality of topological masking may be sensitive to hyperparameters like the number of walks and the stopping probability, making it challenging to generalize the method across different graph structures.

   These limitations should be acknowledged and discussed for a more balanced perspective.

**Questions:**

1. Eq. (4): The power series is generally not guaranteed to converge. It is better to specify clearly the underlying assumptions on W and alpha that guarantee convergence. Is $\alpha_0$ assumed here to equal 1, as in (Reid et al. 2024b)?
2. Remark 3.1:
   - This remark is used as a lemma. Better state it as such.
   - It is in general not guaranteed that alpha has a deconvolution. Is it an assumption of Remark 3.1? Or is it guaranteed by some other assumption? Better clarify.
3. Ln. 115-116: "$\Phi_{Q,K} \in$" should probably be "$\Phi_Q, \Phi_K \in$"
4. Ln. 184-185: Statement is unclear. Why should it necessarily be faster?

---

> ### Author Response · Authors · 2024-11-15
> **Thanks for the review (part 1/2)**
>
> We thank the reviewer for their comments. We are pleased that they find our theoretical contributions valuable, and that they recognise the improvements to predictive performance. We answer all their questions and address points of misunderstanding below.
>
> 1. *Mathematical presentation*. We are sorry that the reviewer found some of the technical exposition difficult to follow. We took great care to include high-level passages (line 224, ‘at a high-level…’) and a visual schematic (Fig. 2), designed to help build intuition. We agree that the notation around Thm 3.1 and the use of McDiarmid’s inequality becomes dense. **Are there any specific sentences or mathematical contributions that the reviewer did not understand, which we may clarify for them?**. We would be very happy to try to rephrase any parts the reviewer can flag as confusing.
> 2. *Limitations*. Respectfully, we do not agree with the reviewer’s list of suggested limitations.
> - *Dependence on graph structure*. One of our key theoretical contributions is that GRF estimators are sharp for *any* graph with bounded $c$ (Thm. 3.1), so it is not true to say that the method depends heavily on graph specifics. This is in sharp contrast to previously published algorithms which are restricted to e.g. grids or trees [1,2]. Intuitively, since we do *importance sampling* of random walks, we still capture long range dependencies. We can sample these long walks by chance, and then upweight them depending on their probability. Please see App. C.2 for detailed ablations. We emphasise that our method works very well for image classification, which may in general require these long range dependencies. This gives evidence that our method captures them.
> - *Dynamic and evolving graphs*. **Our point cloud experiment on page 9 exactly considers this case of a dynamic graph where nearest neighbours must be re-computed at each timestep**. Our method is found to be extremely efficient, and is the best-performing variant compared to the baselines. Indeed, it is not only more accurate, but its training wall clock times are actually **6% faster** compared to the previous SOTA (MP Interlacer baseline). We have added this wall clock time result to the paper. Whilst we agree that computing (approximate) $k$-nearest neighbours on massive point clouds can in general be expensive, this is not a specific of our algorithm. It is a property of this type of dynamic data, and is also required for the MP Interlacer baseline or indeed any GNN-type model in this context.
> - *Choice of hyperparameters*. The reviewer is correct that our method is stochastic, similar to the popular and highly-cited Performers paper [3]. However, **one of our key theoretical contributions is how to choose the number of walkers $n$ and termination probability $p_\textrm{halt}$ to guarantee sharp kernel estimates with high probability** (Thm 3.1). We provide a very specific recipe for choosing them based on our novel results for the estimator concentration; the hyperparameters do *not* need to be manually tuned by the practitioner. Therefore, it is not correct to say that the randomised nature of the algorithm makes it difficult to generalise across graph structures – in fact, the converse is true.
>
> CONTINUED BELOW.

---

> > ### Author Response · Authors · 2024-11-15
> > **Thanks for the review (part 2/2)**
> >
> > *Questions and minor points*
> > 1. *Convergence of the power series and Remark 3.1*. Thanks for the great questions. We refer the reviewer to ‘General Graph Random Features’ [4], the original GRFs paper, for full details. For the reviewer’s convenience, these are summarised as follows.
> > - The power series in Eq. 2 is not in general guaranteed to converge. It converges if $\sum_{i=0}^\infty \alpha_i \lambda^i$ converges for all $\lambda_i \in \Lambda(\mathbf{W})$. In the graph node kernel literature, this is typically ensured by ‘regularising’ or ‘normalising’ $\mathbf{W}$ by taking $\mathbf{W}\to \mathbf{W}/\sqrt{d_i d_j}$ (where $d_i = \sum_j W_{ij}$) to control its spectral radius [4,5]. We describe this in line 168. One also chooses suitable sequence $(\alpha_i)^\infty_{i=0}$ like $\alpha_i=1/i!$ (the heat kernel). $\alpha_0$ does not necessarily need to be assumed to be 1; this just adds an overall scale to the kernel which is not important for predictions.
> > - Eq 6 by Reid et al. [4] shows how to compute $(f_k)^\infty_{k=0}$ from $(\alpha_i)^\infty_{i=0}$, using an iterative formula which can be applied if $\alpha_0>0$. However, of course, this power series is also not guaranteed to converge in general. Once again, this means it is important to control the spectral radius of $\mathbf{W}$ to stay within its radius of convergence.
> > - However, **neither of these details matters for our purposes because, we directly learn $f$** (see line 351). In doing so, we *implicitly* learn the graph node kernel (mask) in feature space [7]. Specifically, during training we learn a sequence of $i_\textrm{max}$ real numbers $(f_i)^{i_\textrm{max}}_{i=0}$. Using a *finite* expansion means the result is guaranteed to converge (and keeps the number of learnable parameters finite), and learning the mask $\mathbf{M}$ in feature space means that 1) we never have to explicitly instantiate it in memory and 2) it is guaranteed to be positive definite. This allows us to elegantly sidestep the problems the reviewer raised above.
> > We again thank the reviewer for raising these important points. We agree that this discussion may be of interest to readers, so have incorporated it into the manuscript.
> >
> > 2. *Notation on line 115*. Thanks for the suggestion. Whilst we chose this notation for compactness, we agree that $\Phi_Q, \Phi_K \in \mathbb{R}^{N \times m}$ may be clearer. Therefore, we have updated it.
> > 3. *Low rank decompositions are fast*. It is well-documented in the literature that the ability to rewrite a kernel as a low rank decomposition is the key to the speed of random feature methods [8]. One ‘stacks’ the features into a design matrix, and uses the associative nature of matrix multiplication to avoid ever instantiating any $\mathcal{O}(N^2)$ object in memory. See Fig. 1 for a visual overview. Line 184-185 summarises this observation, suggesting that for efficient masking one should try to use *graph* random features. The rest of the paper is dedicated to achieving this.
> >
> > We again thank the reviewer for their thoughtful feedback. We believe that we have addressed all their questions and concerns, and have updated the manuscript to incorporate their improvements. We very much hope they will consider raising their score, and warmly invite them to respond.
> >
> > ________________
> > [1] From Block-Toeplitz Matrices to Differential Equations on Graphs: Towards a General Theory for Scalable Masked Transformers, Choromanski et al., ICML 2022, https://doi.org/10.48550/arXiv.2107.07999
> > [2] Stable, Fast and Accurate: Kernelized Attention with Relative Positional Encoding, Luo et al., NeurIPS 2021. URL https://doi.org/10.48550/arXiv.2106.12566
> > [3] Rethinking Attention with Performers, Choromanski et al., ICLR 2021, https://arxiv.org/abs/2009.14794
> > [4] General Graph Random Features, Reid et al., ICLR 2024, https://arxiv.org/abs/2310.04859
> > [5] Kernels and Regularization on Graphs, Smola and Kondor, COLT 2003, https://people.cs.uchicago.edu/~risi/papers/SmolaKondor.pdf
> > [6] Spectral graph theory, Chung, 2007, https://mathweb.ucsd.edu/~fan/research/revised.html
> > [7] Introduction to RKHS, and Some Simple Kernel Algorithms, Gretton, Adv. Top. Mach. Learn. Lecture Conducted from University College London, 16(5-3):2, 2013. https://www.gatsby.ucl.ac.uk/~gretton/coursefiles/lecture4_introToRKHS.pdf
> > [8] Random features for large-scale kernel machines, Rahimi and Recht, NeurIPS 2007 https://people.eecs.berkeley.edu/~brecht/papers/07.rah.rec.nips.pdf

---

> > > ### Comment · Reviewer_BeTo · 2024-11-18
> > > **Response to Author Rebuttal**
> > >
> > > Dear authors,
> > >
> > > > _Mathematical presentation._ We are sorry that the reviewer found some of the technical exposition difficult to follow. [...] **Are there any specific sentences or mathematical contributions that the reviewer did not understand, which we may clarify for them?**
> > >
> > > Thank you for your willingness to improve the clarity of the manuscript. While it is not within my role to provide detailed line-by-line editorial feedback, I recommend revisiting the presentation as a whole to ensure that it is accessible to the reader.
> > >
> > > > _Convergence of the power series and Remark 3.1._ Thanks for the great questions. We refer the reviewer to ‘General Graph Random Features’ [4], the original GRFs paper, for full details. For the reviewer’s convenience, these are summarised as follows.
> > >
> > > These critical details should be explained _within_ the manuscript to ensure comprehensibility for readers who may not have prior familiarity with the work of Reid et al.
> > >
> > > > Eq 6 by Reid et al. [4] shows how to compute [...] However, of course, this power series is also not guaranteed to converge in general. [...] However, **neither of these details matters for our purposes because, we directly learn $f$**. [...] Using a _finite_ expansion means the result is guaranteed to converge.
> > >
> > > Mathematical exposition must be clear and self-contained regardless of later implementation adjustments, as it forms the foundation for the reader’s understanding of the proposed method and validation of the results.
> > >
> > > Based on the current state of the manuscript and the rebuttal, I choose to maintain my score. While I appreciate the authors' efforts to address the review, the concerns outlined above remain unresolved. My score already reflects an acknowledgment of the manuscript's potential while taking into account the weaknesses in its current presentation. I hope the authors will find this feedback useful for further refining the clarity and rigor of their presentation.

---

> > > > ### Author Response · Authors · 2024-11-18
> > > > **Updated manuscript now uploaded: sorry for the confusion, and please take a look**
> > > >
> > > > We apologise to the reviewer for the confusion: at the time of posting our first rebuttal, we were still finalising improvements to the manuscript based on their suggestions. These changes have now been completed and a revision has been uploaded. The latest draft includes **almost a page explaining extra constraints needed to guarantee convergence, and how our method of learning the masking kernel in feature space bypasses them**. Please see Appendix A.1. We have also taken a pass through the text, updating parts which the reviewers flagged as confusing or raised questions.
> > > >
> > > > We believe that we have addressed all the reviewer’s concerns, having added an extra section in the Appendix, updated to the notation, and supplemented with extra explanation. Once again, we respectfully ask them to consider raising their score. We will also be happy to discuss any further suggestions for improvements: of course, we do not expect a line by line editorial, but want the paper to be as impactful as possible so value their suggestions.

---

> > > > > ### Author Response · Authors · 2024-11-24
> > > > > **Any further questions?**
> > > > >
> > > > > As the discussion period draws to a close, this is a polite reminder that we will be happy to answer any further questions the reviewer may have. In particular, we would like to again **draw their attention to the new section explaining the requirements for convergence of graph node kernels and GRFs, and how our method deals with them** (page 16). This was added in response to a concern they raised. We have also clarified minor notational points, explained how our algorithm can be used on arbitrary graphs with bounded $c$, and explained how our algorithm is already found to be efficient with dynamic graphs where nearest neighbours are re-computed on the fly.
> > > > >
> > > > > With the above in mind, we believe that we have addressed all the reviewer’s concerns. We respectfully invite them to confirm whether this is the case and, if satisfied, please consider raising their score. Once again, we thank them for their time and efforts.

---

> > > > > > ### Comment · Reviewer_BeTo · 2024-11-24
> > > > > > **Response to Authors**
> > > > > >
> > > > > > Thank you for your updates. I choose to retain my score until further discussion with the other reviewers.

---

### Official Review · Reviewer_UHQw · 2024-11-03

**Soundness:** 3
**Presentation:** 3
**Contribution:** 3
**Rating:** 8
**Confidence:** 2

**Summary:**

This paper introduces a novel, efficient topological masking approach for transformers on graph-structured data, using learnable functions of the weighted adjacency matrix to adjust attention based on graph structure. By approximating with graph random features, this method supports linear attention, offering strong performance gains across diverse data types and large graphs.

**Strengths:**

1. The prposed method shares $O(n)$ time complexity and suitable for the relatively large scale input.

**Weaknesses:**

see question.

**Questions:**

I appreciate the authors’ valuable contributions in this area. As I am less familiar with applications in image, point cloud, or robotics contexts, I am particularly interested in understanding how Graph Random Features (GRFs) benefit graph neural networks on traditional graph datasets.

1. Could the authors provide examples or case studies that apply GRFs to commonly used graph datasets, such as Cora or Citeseer?
2. Could the authors also include a comparison of computational times between your methods and baseline approaches?

Thanks for this important work, and apologies for my gaps in my background knowledge.  I would also kindly request that the Area Chair reduce the weight of my review in the final evaluation.

---

> ### Author Response · Authors · 2024-11-14
> **Thanks for the review**
>
> We thank the reviewer. We are pleased that, despite having a different research background, they appreciate that our algorithms are novel and scale well to massive datasets. We answer their questions in detail below.
>
> 1. *Applicability of GRFs*. Graph random features were only recently introduced [1,2], so they have not yet been used for the full range of graph-based learning tasks. For instance, **our paper is the first to apply them in Transformers**. However, GRFs have previously been used for efficient kernelised node clustering for Citeseer [3, see Table 2].
> 2. *Computation times c.f. baselines*. We direct the reviewer for Fig. 3, which shows the total number of FLOPs vs. the number of graph nodes for our method (linear + GRFs) compared to the baselines (softmax and unmasked linear). We report FLOPs because it is agnostic to the hardware being used. This experiment confirms that our method is $\mathcal{O}(N)$ time complexity and is faster than full-rank softmax. To supplement with some wall clock times, for the robotics PCT experiment the MP Interlacer baseline [4] (previous SOTA) trains at 0.925 steps/second. Our method, the GRFs Interlacer, trains at 0.982 steps/second. Hence, our method is **not only more accurate, but also faster by 6%**. (The unmasked baseline, which struggles to capture local structure so gives poor accuracy, trains at 1.49 steps/second). We have added these results to the manuscript; thanks for the suggestion.
>
> We again thank the reviewer. Having answered their questions and added some extra wall clock times, we respectfully ask that they consider raising their score.
>
> _________________
> [1] Taming Graph Kernels with Random Features, Choromanski, ICML 2023, https://arxiv.org/abs/2305.00156
> [2] General Graph Random Features, Reid et al., ICLR 2024, https://arxiv.org/abs/2310.04859
> [3] Quasi Monte Carlo Graph Random Features, Reid et al., NeurIPS 2023, https://arxiv.org/abs/2305.12470
> [4 Modelling the Real World with High Density Visual Particle Dynamics, Whitney et al., CoRL 2024, https://arxiv.org/abs/2406.19800

---

> > ### Comment · Reviewer_UHQw · 2024-11-18
> > **Thanks for the authors rebuttal**
> >
> > Thanks for the authors' rebuttal, they address most of my concern, I raise my score to 8, but I cannot raise my confidence due to the lack of background.
> >
> > Regarding Q1, in addition to efficient node clustering, could your method also have other important applications? For instance, could it benefit node classification or other domains currently focused by graph neural networks? This could potentially enhance the contributions of your paper.
> >
> > Good luck.

---

> > ### Comment · Reviewer_UHQw · 2024-11-22
> >
> > Since the public discussion period is about to end, could the author answer my furhter question:
> >
> > "Regarding Q1, in addition to efficient node clustering, could your method also have other important applications? For instance, could it benefit node classification or other domains currently focused by graph neural networks? This could potentially enhance the contributions of your paper."
> >
> > Best

---

> ### Author Response · Authors · 2024-11-22
> **response**
>
> We would like to sincerely thank the Reviewer for the question.
> Efficient linear-attention Transformers with geometrically modulated attention via GRFs can indeed be leveraged in settings, where GNNs are used. This is the case since they can be naturally applied to graph data. In this context, the GRF mechanism serves as a relative positional encoding method, capable of discounting direct interactions between tokens faraway in the metric induced by the particular graph kernel (potentially learnable). For the comparison, in the GNN setting this discounting is usually implemented by modeling direct interactions only between pairs of adjacent nodes.  In fact our application of particle-based dynamics is an example of the usability in the setting where GNNs are used on a regular basis (particle-based dynamics via GNNs is a subject of the voluminous literature). Therefore other potential applications of our methods include in particular bio-informatics (e.g. drug design and molecular biology, where GNNs are machine learning methods of choice).

---

> > ### Comment · Reviewer_UHQw · 2024-11-22
> >
> > Thanks for the answer, I have no question now.

---

### Official Review · Reviewer_Ptpy · 2024-11-04

**Soundness:** 3
**Presentation:** 4
**Contribution:** 3
**Rating:** 6
**Confidence:** 5

**Summary:**

This paper addresses the challenge of incorporating graph structural information into transformer attention mechanisms while maintaining computational efficiency. Their main focus is on topological masking, especially under the low-rank assumption of the attention matrix. The authors use graph random features (GRFs) to approximate topological masks for attention via importance sampling, which are parameterized as learnable functions of the weighted adjacency matrix. They propose a method to control transformer attention using graph node kernels based on random walks via power series of the adjacency matrix, with a random halting probability at each step. They provide concentration bounds in Theorem 3.1. Additionally, their empirical evaluation is carried out on diverse tasks like vision transformers on ImageNet, iNaturalist2021, Places365 and point cloud dynamics prediction for robotics applications.

While the experimental results show good promise, the paper’s theoretical complexity analysis and claims about O(1) GRF sparsity doesn’t hold true for all general graphs. Despite these theoretical issues, the paper introduces interesting ideas about using graph structure in attention mechanisms and provides novel empirical results, particularly in the robotics domain.

**Strengths:**

1. The paper provides strong theoretical foundations with proven concentration bounds and complexity guarantees for GRFs.
2. The method shows concrete performance improvements on real-world tasks and scales to large problems (>30k nodes) that would be intractable with quadratic approaches.
3. The approach can be implemented with both symmetric and asymmetric GRFs, offering different trade-offs between computational efficiency and variance in mask estimation.
4. The experiments cover diverse applications (images, point clouds, videos) and include detailed ablation studies.

**Weaknesses:**

The paper's central claim of O(N) complexity relies critically on the assertion that Graph Random Features (GRFs) have O(1) sparsity. This claim is mathematically incorrect for several reasons:
In Lemma 3.2, while the result doesn’t show an N term, it is still implicitly dependent on the size of the graph. O(1) complexity implies that your non-zero entries per row vector $\hat{\phi}G(vi)$ are bounded by a constant independent of input size. The bound in Lemma 3.2 is still dependent on multiple parameters like $n$, $p_halt$ and $\delta$. So, one can say that your complexity is like the complexity of a “parameterized algorithm”, i.e., $O(f(n,p_halt,\delta))$, where f is some function of the parameters.
Let's consider a family of complete graphs ${G_N}{N \geq 1}$ where $G_N$ has N vertices and each vertex has degree N-1. Then all edge weights are equal, i.e., $1/ sqrt( (N-1)(N-1) )$.
At any step, the walk can move to another vertex with probability 1/(N-1).
For a given walk starting at an arbitrary vertex, if you assign a r.v. to count the number of “distinct vertices” visited, even with the inclusion of geometric termination, you will find that this r.v grows with N because it has (i) more possible vertices available at each step to visit, (ii) the probability of visiting a new vertex at each step increases with N, and (iii) each successful step before halting can easily reach O(N) vertices. Therefore, making O(N) non-zero entries in the row vector and hence your attention matrix. With more independent walks starting from v, you can fill up even more non-zero entries. Hence, the O(1) bound doesn’t hold here.

As a demonstrative simple counter-example, consider the following two cases with fixed parameters. Let’s fix the parameters as n=10 (walks), p_halt = 0.5 and \delta=0.1

Case 1: Small complete graph
- N = 10 nodes
- Each node has degree 9
- Even a 1-hop walk can reach 9 other nodes
- A 2-hop walk can reach all nodes

Case 2: Larger complete graph
- N = 1000 nodes
- Each node has degree 999
- A 1-hop walk can reach 999 other nodes
- A 2-hop walk can reach all nodes

While their bound might be the same in both cases, as it depends only on $n$, $p_halt$ and $\delta$, but the number of non-zero entries in both cases ends up being very different from one another. In the second case, you are much more likely to get O(N) non-zero entries due to the reasons I mentioned earlier about each hop having many more options, more distinct nodes, thus more reachability and coverage of the underlying graph (or at least exploring a large portion of the graph before terminating).

This demonstrates that actual sparsity heavily depends on the structure of the graph.

This rigorous analysis shows that the number of non-zero entries cannot be independent of graph size without additional constraints on the graph structure. The analysis shows that the results proposed by the author hold only with some assumptions on the graph structure, for example sparse graphs of bounded-degree graphs.

The authors make a significant assumption in their discussion of Theorem 3.1. (Lines 278-280), where they state that "assuming that c remains constant (i.e. we fix a maximum edge weight and node degree as the graph grows)..."
This reveals that their theoretical analysis works only for graphs with a bounded maximum degree and hence contradicts the claims about working for general graphs (line 82 in Introduction). This algorithm cannot handle dense graphs, complete graphs (or almost complete graphs), graphs where node degrees grow with N and many real world graphs where there can be very high-degree nodes and no bounds on degrees.

The authors should make this bounded-degree assumption explicit upfront in the introduction and modify their claims about general graphs.

A complete graph (or almost complete graph) isn’t completely unusual especially in the context of attention mechanisms where in the final layers pretty much end up having all tokens attending to each other in a pairwise manner. The experiments done in the paper are done on very low-degree graphs like grid graphs, which doesn’t demonstrate the applicability of their method to general graphs, especially large dense ones.

Inconsistent and confusing notation:
- $(f_k)_{k=0}^\infty$ is sometimes treated as a sequence of reals, sometimes as a function
- Incorrect set notation: claiming (f_k)_{k=0}^∞ ⊂ R when sequences are functions from N to R
- Weighted adjacency matrix definition issues:
  * Claim W is weighted but then suggest $w_{ij} = 1/sqrt(d_i d_j)$
  * This normalization discards meaningful edge weights in attention context

ii) Missing crucial assumptions:
- No explicit assumptions about graph structure
- No discussion of how graph density affects complexity
- No proper analysis of how maximum degree impacts sparsity

The paper's main contribution is focused specifically on making topological masking efficient, rather than improving linear attention in general. The paper makes empirical comparisons to only basic linear attention models and focuses solely on topological masking efficiency.

In the experiments section, it would be interesting to compare against other major linear attention variants like (i) Performers (Choromanski et al., 2020) Favor+ which uses kernel approximations and (iii) Nyströmformer which uses Nystrom’s method to approximate attention.

**Questions:**

While the empirical results might be interesting, the fundamental theoretical claims that form the paper's main contribution are incorrect. A major revision would be needed to:
1. Correct the theoretical analysis
2. Properly characterize complexity and mention which category of graphs it can address
3. Either prove better bounds under specific assumptions or acknowledge limitations
4. Frame results in terms of parameterized complexity

The authors should consider addressing these fundamental issues during rebuttal, if possible.

---

> ### Author Response · Authors · 2024-11-13
> **Thanks for the review (1/2)**
>
> (Part 1/2)
>
> We thank the reviewer for their comments. We are pleased that they note our work’s strong theoretical foundations, concrete performance improvements, diverse applications and ablation studies. We are grateful for their time.
>
> However, we **strongly disagree that any of the theoretical analysis in the paper is incorrect**. We will clarify points of misunderstanding below.
>
> To recapitulate the results in the paper, our key theoretical contributions are as follows.
> 1. Theorem 3.1 gives the first known concentration bound for GRFs. The bound depends on the number of walkers $n$, and a scalar denoted $c$. As the reviewer notes, $c$ depends on the underlying graph via its maximum node degree and edge weight: specifically, via $\max_{v_i, v_j \in \mathcal{N}}(|\mathbf{W}_{ij}| d_i)$ (line 266).
> 2. Meanwhile, Lemma 3.2 bounds the GRF sparsity for *any* graph, given some number of walkers $n$ and termination probability $p_\textrm{halt}$. The simple proof is based on bounding the number of hops a walker can take before it terminates, supposing its length is geometrically distributed. For given $n$ and $p_\textrm{halt}$, this bound is independent of $\mathcal{G}$ (though of course how tight it is depends on the structure of $\mathcal{G}$; it is generally tighter for denser, bigger graphs).
> 3. Corollary 3.3 combines the two observations above to show that, *provided $c$ remains constant as the graph grows* – meaning we do not have to update our bound so we can use the same number of walkers $n$ and termination probability $p_\textrm{halt}$ without compromising estimator accuracy – then the number of nonzero GRF entries is bounded by a constant with high probability when $N$ gets large. As the reviewer correctly notes, $\mathcal{O}(N)$ time complexity follows.
>
> To be clear, Theorem 3.1 and Lemma 3.2 hold for any graph. Corollary 3.3 does indeed require that $c$ remains constant as the graph grows, in order that the same bound holds so we can safely use the same $p_\textrm{halt}$ and $n$. **We are totally upfront about this requirement**. In Sec. 3.4, we emphasise it **three times**, including in the core statement of the corollary: see lines 278, 295 and 313. We initially omitted this technical detail from the introduction, but on reflection agree that the interested reader might benefit from encountering it earlier in the text. Therefore, we have now updated the paper introduction to flag it even more explicitly – thanks for the suggestion.
>
> *$c$ remaining constant as $N$ grows is not very restrictive.* Though sparsity can be used to ensure that $c$ remains constant as $N$ grows, **this is not a necessary condition**. For instance, in the reviewer’s example of a complete graph with edge weights $1/(N-1)$ , we have that $\max_{v_i, v_j \in \mathcal{N}}(|\mathbf{W}_{ij}| d_i) = 1$  which is independent of $N$. This is an example of a dense graph for which our assumption about $c$ holds, so it is not the case that the analysis only holds for sparse graphs. Stepping back to take a broader perspective, the reason we need to control $c$ is to prevent the spectral radius of the adjacency matrix $\mathbf{W}$ diverging as the graph becomes large. If we do not do this, the underlying exact kernel (defined in Eq. 4) will also diverge, in which case one clearly cannot approximate it with GRFs or otherwise.
>  'Regularising’ or 'normalising’ $\mathbf{W}$ to control its spectral radius is very standard in the graph node kernel literature [1,2,3]. It is not a weakness of our specific approach to topological masking.
>
> *Regularising W*. To build on the above, one typically 'regularises' by taking $W_{ij} \to W_{ij}/ \sqrt{d_i d_j}$  with $d_i = \sum_j W_{ij}$. This bounds its spectral radius to $1$ [2]. Since we start with unweighted adjacency matrices this gives edge weights $1/\sqrt{d_i d_j}$, but our method **does not formally require this**.
>
> *The reviewer’s proposed counterexample does not consider asymptotic $N$*. `Big O’ notation describes the limiting behaviour of the time complexity when $N$ tends to infinity. In contrast, the reviewer’s proposed counterexample looks at the small $N$ regime, where any asymptotic analysis inevitably breaks down. To make this concrete, consider a complete graph with $n=1$ walks and termination probability $p_\textrm{halt}$. When the number of nodes $N$ becomes very large, the probability of a walker backtracking becomes small, so at every timestep it hops to a new, unvisited node. The walker length will be $\frac{1}{p_\textrm{halt}}$ on average, so at asymptotic $N$ only $\sim \frac{1}{p_\textrm{halt}}$ coordinates of the GRF will be nonzero. This is manifestly independent of $N$. In contrast, the reviewer’s example considers the *small* $N$ regime, where time complexity scaling results are not expected to hold. In experiments, this behaviour is shown by the small nonlinear regime at the far left of Fig. 3. There is no problem with our theoretical claims.
>
> CONTINUED BELOW.

---

> ### Author Response · Authors · 2024-11-13
> **Thanks for the review (2/2)**
>
> (Part 2/2)
>
> *Focus on topological masking c.f. improving linear attention.* The reviewer is correct that our paper focuses on developing new topological masking (graph RPE) techniques that are compatible with existing linear attention algorithms. Our methods are agnostic to the particular feature map  $\phi$ used to replace softmax in linear attention: optimising $\phi$ is not the goal of the work. However, we agree that it may be interesting to see the gains that topological masking provides to other linear attention variants, so we are running **extra experiments with positive random features [FAVOR+, 4] instead of ReLU**. Preliminary results are already complete and, as expected, we again see strong gains from incorporating GRFs. For ViT-B/16 trained from scratch on ImageNet with FAVOR+ attention ($m=256$ random features), we see a relative improvement of **+2.3%** from our algorithm compared to the unmasked baseline ($p=0.1$, $n=100$ walks). Once the rest of the additional experiments are complete, we will add them to the manuscript. Thanks for suggesting this.
>
> *Minor points*:
> 1. $f: \mathbb{N} \to \mathbb{R}$ can be interpreted as a function mapping from the natural numbers to real numbers. $(f_k)^\infty_{k=0}$ is a sequence of reals, intended to denote the evaluations of some particular $f$ for the natural numbers rather than a sequence of functions. We will make this clear. Thanks for the suggestion.
> 2. Normalisation of $\mathbf{W}$. Please see earlier comments. This is standard practice to ensure that the kernel converges.
>
> We again thank the reviewer for their time. Having clarified misunderstandings about our theoretical contributions and added extra experiments with the FAVOR+ linear attention mechanism, we hope that they will raise their score. We warmly invite them to respond with any further questions.
>
> __________
> [1] Kernels and regularization on graphs, Smola and Kondor, COLT 2003, https://people.cs.uchicago.edu/~risi/papers/SmolaKondor.pdf
> [2] Spectral graph theory, Chung, 2007, https://mathweb.ucsd.edu/~fan/research/revised.html
> [3] General graph random features, Reid et al., ICLR 2024, https://arxiv.org/pdf/2310.04859
> [4] Rethinking attention with Performers, Choromanski et al., ICLR 2021, https://arxiv.org/pdf/2009.14794

---

> > ### Comment · Reviewer_Ptpy · 2024-11-17
> > **Reponse to 1st rebuttal**
> >
> > Thank you for your detailed response. I appreciate the clarification regarding the theoretical analysis, particularly concerning complete graphs and the handling of asymptotic behavior.
> > I acknowledge that when treating $n$ and $p_{halt}$ as constants, your $O(N)$ complexity claims and theoretical bounds hold. My concern was mostly that, your analysis of the $n=1$ case for complete graphs, provides an interesting insight - as $N \rightarrow \infty$, the probability of backtracking becomes negligible and the walker tends to visit new nodes at each step until termination. While this certainly gives ~$1/p_{halt}$ nonzeros for a single walker, this behavior actually highlights a potential practical concern:
> >
> > With $n$ independent walkers, each walker will likely visit different nodes (since probability of hitting previously unvisited nodes remains high in large complete graphs).
> > This suggests that in practice, the total number of unique nodes visited (and thus nonzeros) will approach $O(min(n/p_{halt}, N))$. While this is still $O(1)$ when treating $n$ and $p_{halt}$ as constants (as you do), it indicates that:
> >
> > - The sparsity bound might not be so tight in practice for dense graphs
> > - The actual number of nonzeros could be substantial when using enough walkers for good approximation
> > - There is a tradeoff between approximation quality (which requires larger $n$) and achieved sparsity
> >
> > Could you provide an empirical analysis of how the number of unique nodes visited scales with $n$ for dense graphs? This would help practitioners understand the practical implications of this behavior.
> >
> >
> > 1. **Sparsity vs Approximation Quality:**
> > While each GRF vector has $O(1)$ nonzeros for fixed parameters, how many walkers are actually needed to maintain a good approximation quality? In practice, might $n$ need to scale with graph size/density to achieve desired accuracy?
> > Could you provide some empirical analysis showing how sparsity patterns change with different $n$ values across graph sizes?
> >
> > 2. **Practical Considerations:**
> >  Your experiments focus on sparse graphs (grids, point clouds). How does the method perform on denser graphs in practice?
> >  Could you please provide a more detailed empirical analysis of the relationship between: the number of walkers ($n$), the halting probability ($p_{halt}$), achieved sparsity, approximation accuracy, and the computational cost?
> > This would help practitioners understand the practical implications of the theoretical sparsity bounds.
> >
> > 3. Can you please discuss what are the practical limitations of using fixed parameters for very large or dense graphs?
> >
> > I understand that this is extra work, but I believe this will help a lot. Given your clear theoretical explanation, I would be happy to raise my score if the paper is revised to include empirical analysis addressing the practical concerns raised above, particularly regarding scaling behavior with multiple walkers on dense graphs. This would provide valuable guidance for practitioners while complementing your strong theoretical results.

---

> ### Author Response · Authors · 2024-11-17
> **Thanks for the further response**
>
> We thank the reviewer for their quick reply. We are grateful for their perspective
>
> The reviewer is correct to note that, for a given $p_\textrm{halt}$ and number of walkers $n$ (selected to guarantee sharp kernel estimates by Thm. 3.1), the actual sparsity of the corresponding GRFs can depend on the particular graph being considered. For sparse graphs GRFs will tend to be more sparse (since there is a higher probability of backtracking with fewer neighbours), and for dense graphs GRFs will tend to be less sparse (because walkers are more likely to visit a new, previously unvisited node). However, **the bound in Lemma 3.2 already considers the worst possible case, when walkers never backtrack**. You can see this in lines 288 and 289, when we say *‘$n$ walkers are all $b$ or shorter with probability $(1-(1-p_\textrm{halt})^b)^n$... at most $bn$ entries of $\widehat{\phi}(v_i)$ can then be nonzero’*. Clearly, this corresponds exactly to the scenario when the number of visited nodes is equal to the total number of hops. Therefore, our bound will actually be **tighter for denser graphs**. Put simply, our algorithm will be *even more efficient* for sparse graphs, whereas for dense graphs the time complexity will be closer to the theoretical worst case. Of course, it is $\mathcal{O}(N)$ in both scenarios.
>
> We agree that the question of how GRF sparsity and mask (kernel) approximation quality are related is an interesting one. We direct the reviewer to ‘General Graph Random Features’ [1], the paper which introduced the algorithm, which provides **a detailed investigation of this question in App A.6**. Respectfully, this kind of ablation for GRFs is not the main focus of this paper, which instead provides new concentration bounds and applies them to a range of tasks in Transformers. Nonetheless, we do agree that the interested reader may benefit from some brief empirical investigation to avoid needing to separately refer to this previous work.
>
> For this reason, following the reviewer’s suggestion, we have now added a **new experiment to the Appendix – please see App. E**. Taking different Erdős–Rényi graphs with edge probabilities between $0.1$ and $0.9$, we show how 1) the mask approximation quality and 2) the GRF sparsity varies. As anticipated, both become slightly worse then plateau as the graphs become denser. However, they vary within a narrow range of $\sim$ 10%, so this need not be a big practical concern. We also provide similar plots for a varying number of graph nodes $N$ with fixed graph sparsity (edge probability 0.5), where we again find that GRF performance tends to be a little better for smaller graphs but quickly plateaus for big graphs. Meanwhile, the GRF sparsity drops as $N$ grows: at most $\mathcal{O}(1)$ entries are nonzero, so the proportion of nonzero entries goes down as $\mathcal{O}(1/N)$. **We emphasise once again that these experiments serve to show how *tight* the bounds are, but the bounds themselves still hold in every case – put simply, one may get an even faster algorithm for very sparse, small examples, but our theoretical results prove that the algorithm is efficient in all scenarios**. Note especially that in every case the mask approximation quality remains excellent, characterised by a tiny relative Frobenius norm error, even for dense, big topologies.
>
> **We thank the reviewer for prompting us to add these experiments regarding scaling behaviour with multiple walkers on dense graphs to the paper. We trust that they have allayed any remaining practical concerns**. We also politely remind them that our Transformer experiments already include **three** different data modalities across multiple datasets and tasks: images, videos and point cloud data for robotics. Previously proposed algorithms for topological masking for special graphs (sequences, grids or trees) have typically focussed on just one [2,3]. As such, we respectfully suggest that we have already shown that our algorithm is very practical in a broad range of settings. Nonetheless, we agree that these latest additions have improved the paper – thanks for the suggestions.
>
> With these additions and clarifications (of course, as well as the additional FAVOR+ experiments we are running for the reviewer), would the reviewer please consider raising their score and recommending acceptance?
>
> _______
> [1] General Graph Random Features, Reid et al., ICLR 2024, https://arxiv.org/abs/2310.04859
> [2] From Block-Toeplitz Matrices to Differential Equations on Graphs: Towards a General Theory for Scalable Masked Transformers, Choromanski et al., ICML 2022, https://doi.org/10.48550/arXiv.2107.07999.
> [3] Stable, Fast and Accurate: Kernelized Attention with Relative Positional Encoding, Luo et al., NeurIPS 2021. URL https://doi.org/10.48550/arXiv.2106.12566

---

> > ### Comment · Reviewer_Ptpy · 2024-11-17
> > **Reply**
> >
> > Thanks for adding the extra experiments. I have revised my initial score from 3 (reject) to 6 (weak accept) based on your further analysis. However, I would like to reserve making this my final score until after the reviewer discussion period.

---

> > > ### Author Response · Authors · 2024-11-17
> > > **Anything else?**
> > >
> > > We thank the reviewer for their response, and for raising their score. We are pleased that they agree that our theoretical analysis is correct and that our extra experiments (GRFs ablations and FAVOR+ attention) have allayed their concerns about the algorithm's practicality.
> > >
> > > Is there anything else they would like to see, or any other questions we can clarify, in order to further increase the score to 'accept, good paper'? We will be happy to discuss further.

---

### Author Response · Authors · 2024-11-27
**Overall response: thanks for the reviews**

We thank the reviewers for their efforts. **We are pleased that, following clarification of minor points of misunderstanding and manuscript updates, all concerns appear resolved and all reviewers recommend acceptance**. To summarise our improvements (shown in red):

1. Notational tweaks, correction of typos, and rephrasing of confusing passages
2. Extra technical discussion about convergence (App A.1; not directly relevant for our work but perhaps helpful context for readers)
3. Wall-clock times for the Interlacer experiment (line 482)
4. A few additional ablations (time complexity scaling vs. number of walkers, FAVOR+ attention instead of ReLU, GRF approximation quality and sparsity vs. graph sparsity and size; pages 21 and 25)

Once more, we thank the reviewers and the AC. Of course, we will be very happy to answer any new questions during the remainder of the discussion period.

---

### Meta-Review · Area_Chair_BBQt · 2024-12-19

**Metareview:**

The authors consider the transformer attention with graph structured data. In particular, the authors focus on the topological masking of low-rank attention. The authors propose to leverage the graph random features to approximate topological masks and parameterize it as a learnable function of a weighted adjacency matrix. The authors also derive the concentration bounds and show the linear time and space complexity for the proposed approach. The authors illustrate its empirical advantages on images and point cloud data. All Reviewers agree that it is a good submission for ICLR'2025. We urge the Authors to incorporate the Reviewers' comments and discussion in the rebuttal to the updated version, especially regarding time consumption.

**Additional Comments On Reviewer Discussion:**

The authors clarify its linear complexity w.r.t. the number of tokens and its dependence on other parameters. However, the advantage on the time consumption seems too marginal and blurry.

---

### Decision · Program_Chairs · 2025-01-22

Accept (Poster)